# Human respiratory organoids sustained reproducible propagation of human rhinovirus C and elucidation of virus-host interaction

Cun Li [1,10], Yifei Yu[1,10], Zhixin Wan[1], Man Chun Chiu [1,2], Jingjing Huang[1,2], Shuxin Zhang[1], Xiaoxin Zhu[1,2], Qiaoshuai Lan[1,2], Yanlin Deng [1,2], Ying Zhou[1], Wei Xue[1], Ming Yue [1], Jian-Piao Cai[1], Cyril Chik-Yan Yip[1], Kenneth Kak-Yuen Wong [3], Xiaojuan Liu [4], Yang Yu[4], Lin Huang[5], Hin Chu [1,2,6], Jasper Fuk-Woo Chan [1,2,6,7], Hans Clevers [8,9], Kwok Yung Yuen [1,2,6,7] & Jie Zhou [1,2,5,6] ✉

The lack of a robust system to reproducibly propagate HRV-C, a family of viruses refractory to cultivation in standard cell lines, has substantially hindered our understanding of this common respiratory pathogen. We sought to develop an organoid-based system to reproducibly propagate HRV-C, and characterize virus-host interaction using respiratory organoids. We demonstrate that airway organoids sustain serial virus passage with the aid of CYT387-mediated immunosuppression, whereas nasal organoids that more closely simulate the upper airway achieve this without any intervention. Nasal organoids are more susceptible to HRV-C than airway organoids. Intriguingly, upon HRV-C infection, we observe an innate immune response that is stronger in airway organoids than in nasal organoids, which is reproduced in a Poly(I:C) stimulation assay. Treatment with α-CDHR3 and antivirals significantly reduces HRV-C viral growth in airway and nasal organoids. Additionally, an organoid-based immunofluorescence assay is established to titrate HRV-C infectious particles. Collectively, we develop an organoid-based system to reproducibly propagate the poorly cultivable HRV-C, followed by a comprehensive characterization of HRV-C infection and innate immunity in physiologically active respiratory organoids. The organoid-based HRV-C infection model can be extended for developing antiviral strategies. More importantly, our study has opened an avenue for propagating and studying other uncultivable human and animal viruses.

Human rhinovirus (HRV) is a nonenveloped, positive-stranded RNA virus of the Piconaviridae family and the Enterovirus genus, grouped into three species (HRV-A, HRV-B, and HRV-C)[1]. HRV is the most common pathogen of acute upper respiratory infection, linked with lower respiratory infections and exacerbation of asthma and chronic pulmonary diseases. HRV-A and HRV-B are composed of around 100 serotypes and can be readily propagated in immortalized cell lines, using intercellular adhesion molecule 1 (ICAM-1) and low-density

lipoprotein receptor (LDLR) for cellular entry. Since HRV-C was identified in 2006[2], over 60 distinct HRV-C sequences have been reported so far by molecular approaches[3]. In contrast to HRV-A and HRV-B, HRV-C is unable to infect and replicate in standard cell lines. Tremendous efforts have been engaged to tackle the problem during the past 2 decades. Bochkov et al. reported the first in vitro HRV-C infection in primary tissues from surgically resected sinus mucosa and nasal polyps, and they successfully used these tissues to propagate a single HRV-C15 strain. However, virus growth exhibited significant variations, which may be related to the tissues from different donors, and variable status of tissues[4]. Using primary tissues for virus cultivation has presented several challenges, including limited accessibility of the tissues, the difficulty of standardization, and tissue degradation during experimentations[5]. Subsequently, the same team identified CDHR3 as the cellular receptor of HRV-C[6], a landmark discovery in HRV-C research.

HRV-C infection and limited propagation were also reported in primary human airway epithelial cells[7,8]. Air-liquid interface cultures of the primary human airway and nasal epithelial cells allow mucociliary differentiation into the pseudostratified ciliated epithelium. These primary epithelial cells have been used to delineate respiratory viral infections, including COVID-19[9,10]. Yet the limited expansion capacity inherent to primary epithelial cells substantially restricts their application for routine experimentations, including virus isolation and serial virus passage. Moreover, HRV-C is not the sole uncultivable virus. Other human viruses of respiratory tropism, such as bocavirus[11] and human coronavirus HKU1[12], are unable to infect immortalized cell lines, except in ex vivo respiratory tissues and primary airway epithelial cells; these human viruses have yet to be cultivated for comprehensive investigations. Therefore, there is an unmet need for a robust and readily accessible cultivation system to propagate these uncultivable viruses, which is the initial step for understanding these pathogenic microbes. Overall, the lack of a robust cultivation system represents a major hurdle to understanding the biology of HRV-C and other uncultivable respiratory viruses, their interaction with human respiratory cells, and the development of antiviral strategies against these common respiratory pathogens.

Organoids, also called "mini-organs," are three-dimensional (3D) cellular clusters derived from stem cells, including embryonic stem cells, induced pluripotent stem cells, and organ-specific adult stem cells in primary tissues. They can self-renew and self-organize into complex, functional structures that recapitulate the cellular diversity and physiological functions of corresponding tissues and organs. We have established the first human respiratory organoid culture system[13–19]. Organoids are derived from adult stem cells in primary lung tissues with high efficiency and can be stably expanded over half a year. We have developed differentiation protocols to induce maturation in long-term expandable organoids and generate mature airway organoids and alveolar organoids that faithfully simulate the native airway and alveolar epithelium, respectively[16]. Apart from a non-invasive procedure to procure readily accessible nasal cells for organoid derivation, nasal organoids more adequately simulate the upper airway epithelium than airway organoids[17]. In this two-phase organoid culture system[16,17], expansion culture provides a stable and long-term expanding source, while differentiation protocols enable us to generate large amounts of physiologically active respiratory epithelial cells (Illustrated in Fig. 1a). Thus, the respiratory organoid culture system allows us to rebuild and propagate the entire human respiratory epithelium in culture plates with excellent efficiency and stability. These respiratory organoids have become robust and popular tools for studying SARS-CoV-2 and other respiratory viruses[16,17,20–22].

Given the respiratory tropism of HRV-C and the ability of nasal and airway organoids to accurately simulate the native epithelium in human airways, we hypothesized that these respiratory organoids (refer to airway and nasal organoids hereafter) may be susceptible to

HRV-C and sustain virus cultivation. We were prompted to develop an organoid-based culture system to reproducibly propagate HRV-C, and characterize HRV-C infection and complex virus-host interaction in these respiratory organoids.

## Results

### Human airway organoids susceptible to HRV-C positive clinical specimens

A total of nine achieved HRV-C positive nasopharyngeal specimens with variable viral loads were obtained. We inoculated these specimens onto the 2D human airway organoids, monolayers of airway organoids growing on transwell inserts. We harvested an aliquot of culture medium from the top chamber on the indicated days post-inoculation to detect viral load (copy number of viral 5′UTR) over time. As shown in Fig. 1b, 8 out of 9 specimens showed an increased viral load after inoculation, indicating a productive HRV-C infection of the airway organoids. Phylogenic analysis of the 5′UTR of the eight specimens revealed that specimens 1, 4, 5, and 7 were closely related to HRV-C15, while the others were related to HRV-C3, -C8, -C11, and -C45 (Supplementary Fig. 1). Phylogenetic analysis based on the VP4/VP2 genes, which are also commonly used for rhinovirus genotyping, showed similar results (Supplementary Fig. 2). However, specimen 6 was genotyped to be HRV-C45 and HRV-C11 based on the 5′UTR and VP4/VP2 sequences, respectively. The inconsistency was also reported in HRV genotyping previously[23]. We then examined HRV-C viral growth and release in airway organoids. After the inoculation of an HRV-C3 positive (HRV-C3 + ) specimen, we harvested culture media from the top and bottom chambers for viral load detection. As shown in Fig. 1c, viral loads were remarkably higher in the apical media than in the basolateral media, suggesting that progeny virions were preferentially released from the apical side. HRV-A was inoculated onto the airway organoids in parallel. We observed a productive infection of HRV-A with a similar pattern of virus release (Fig. 1c).

Understanding the biology of a virus invariably starts with reproducible virus isolation and cultivating a sufficient amount of viruses, followed by a series of in vitro and in vivo characterizations of virus biology and virus-host interaction. Given that airway organoids were susceptible to HRV-C infection and sustained active viral replication, we inferred that airway organoids might sustain serial virus passage, enabling reproducible HRV-C isolation and cultivation. We were prompted to propagate the progeny virions using the culture media from the first round of infection with HRV-C+ clinical specimens. However, when the HRV-C+ media were used to inoculate a new batch of airway organoids, we did not observe any sign of viral growth. Namely, airway organoids, despite the susceptibility to HRV-C, were unable to sustain viral serial passage.

### Immunosuppression enabling serial HRV-C passage in airway organoids

Prior studies in experimental mice and human epithelial organoids documented that mucosal epithelial cells elicited a robust antiviral response upon viral infections[24–28]. We inferred that the antiviral response triggered in airway organoids by HRV-C infection might restrain serial viral passage. CYT387, a drug for the treatment of myelofibrosis, can inhibit interferon and interferon-stimulated gene expression through JAK1/JAK2 (kinases mediating IFN signaling) and TBK1/IKKε (kinases mediating RLR and TLR signaling) pathways[27]. A clinical study reported that JAK1/JAK2 inhibition rendered patients at a higher risk of rhinovirus infection[29]. Therefore, we hypothesized that CYT387 treatment may repress the virus-induced antiviral response in airway organoids, boost viral propagation, and enable serial HRV-C passage.

We examined HRV-C replication in airway organoids in the presence or absence of CYT387 (Fig. 2a). Briefly, we inoculated an HRV-C3+ medium (medium collected from organoids infected by an HRV-

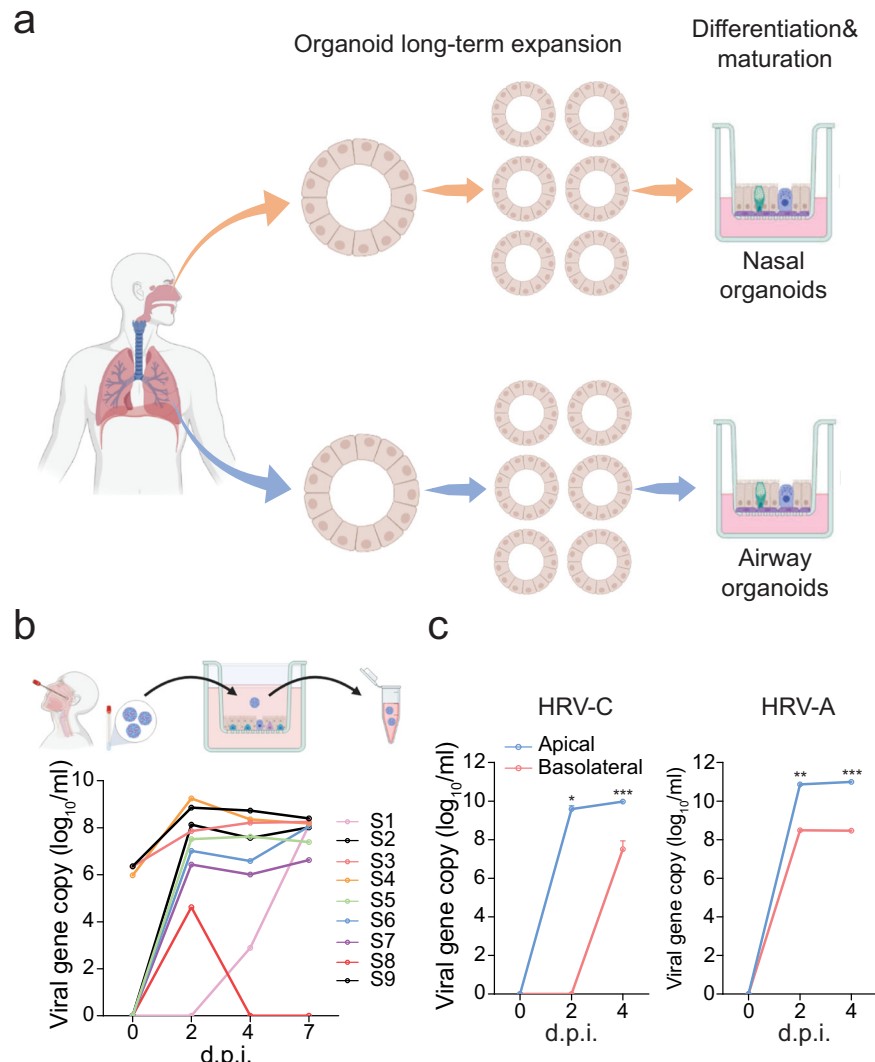

**Fig. 1 | Human airway organoids were susceptible to clinical specimens of HRV-C. a** A schematic illustration of the human nasal organoid and airway organoid culture system was created with Biorender.com. **b** Airway organoids (in 9 transwell inserts) were inoculated with 9 HRV-C+ nasopharyngeal aspirates. At the indicated day post-infection (d.p.i.), culture media were harvested from the infected airway organoids and applied to viral load detection by RT-qPCR. A schematic graph of the experimental procedure was created with Biorender.com. **c** Airway organoids were inoculated with HRV-A1 and HRV-C3 at 100 viral gene copy/cell ($n = 3$). Culture media were harvested from apical and basolateral chambers of infected airway organoids at the indicated d.p.i. and applied to the viral load detection. Data represent mean and SD of the indicated number (n) of biological replicates from a representative experiment independently performed three times. Statistical significance (in panel c) was determined using a two-tailed Student's t-test. **$P < 0.01$, ***$P < 0.001$. Source data are provided as a Source Data file.

C3+ clinical specimen) onto the airway organoids pre-treated with 1 µg/ml CYT387 or DMSO. The infected organoids were further incubated with CYT387 (CYT thereafter) or DMSO for 96 h, after which we harvested the culture medium (P1) to examine viral growth. P1 media from CYT- and mock-treated organoids were brought forward to P2 passage in the presence of CYT and DMSO, respectively (Fig. 2a). As shown in Fig. 2b, CYT treatment resulted in an elevated viral load in P1 media at 96 h post-infection (h.p.i.). In P2, CYT-treated organoids still sustained robust viral replication, whilst no viral growth was observed in DMSO-treated organoids. In the following P3 passage, we had to inoculate CYT-treated P2 media onto airway organoids, in which we continued CYT or DMSO treatment to verify the enhancement effect of CYT. This CYT-treated media inoculation, followed by CYT or DMSO treatment, was repeated in the subsequent P4 and P5 passage. In the presence of CYT, HRV-C3 was consecutively passaged five times in airway organoids, with a higher viral load than that in DMSO-treated organoids (Fig. 2c). Overall, the results indicated that airway organoids enabled serial HRV-C propagation in the presence of CYT. Moreover, HRV-C virions serially propagated in CYT-treated organoids

established a productive infection in airway organoids without CYT treatment, indicating that progeny virions passaged in airway organoids in the presence of CYT were highly infectious (Fig. 2d, Supplementary Fig. 3).

## Nasal organoids sustaining serial HRV-C propagation
We then performed virus passage in nasal organoids, the organoid model with a cellular composition comparable to airway organoids, yet better resembles the upper respiratory epithelium[17]. We inoculated nasal organoids with the HRV-C3+ medium collected from organoids infected with an HRV-C3+ clinical specimen in the presence of CYT or DMSO and collected culture media at 96 h.p.i. Due to the high viral load in DMSO-treated nasal organoids in a preliminary experiment, we changed the experimental design. Briefly, the virus-containing media from CYT- and DMSO-treated organoids were brought forward to the next round of infection with the CYT and DMSO treatment, respectively (Fig. 3a). Notably, CYT treatment significantly boosted viral growth (Fig. 3b). Nonetheless, in the absence of CYT, nasal organoids derived from two different donors sustained four consecutive

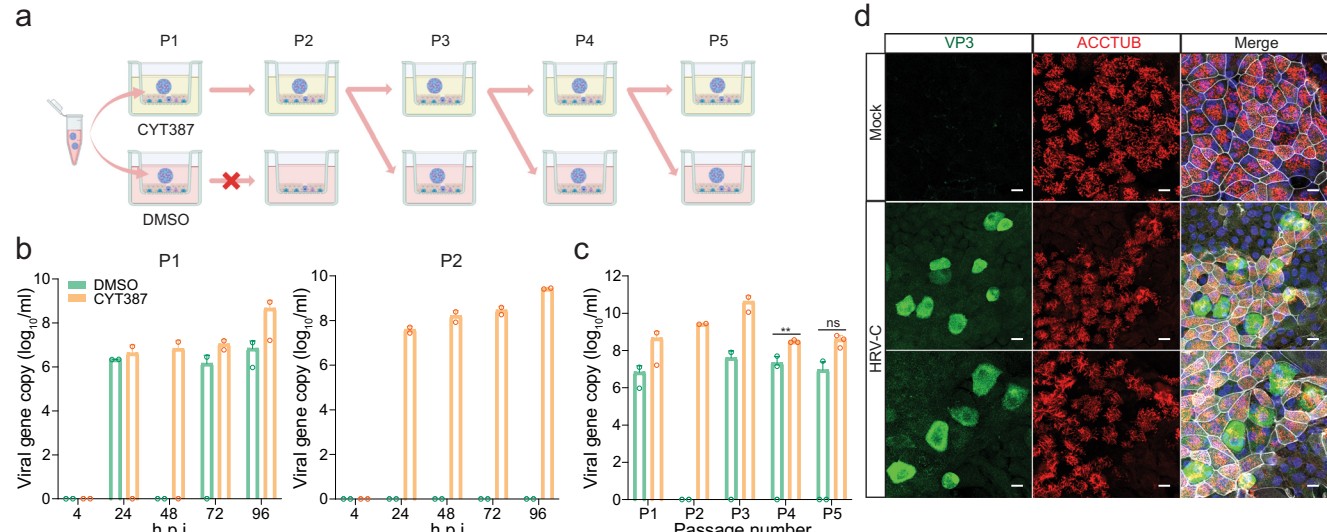

**Fig. 2 | CYT387 enabled serial propagation of HRV-C in human airway organoids. a** A schematic graph outlines the experimental procedure for panels (**b** and **c**), created with Biorender.com. **b** In the first (P1) passage, after pre-treatment with CYT387 or DMSO overnight, airway organoids were inoculated with HRV-C3 (*n* = 2). Culture media were harvested from infected airway organoids at indicated h.p.i. to detect viral replication. In the second (P2) passage, airway organoids pretreated with CYT387 or DMSO were inoculated with 50 μl P1 medium collected from CYT387- or DMSO-treated organoids, respectively (*n* = 2). Culture media were harvested at indicated h.p.i. to detect viral replication. **c** From P3, airway organoids pretreated with CYT387 or DMSO were inoculated with medium collected from CYT387-treated organoids at 100 viral gene copy/cell (P3, *n* = 2; P4 and P5, *n* = 3).

The culture media were harvested at 96 h.p.i. to detect viral replication. The viral loads in P1 and P2 media at 96 h.p.i. are incorporated. Data represent mean and SD of the indicated number (n) of biological replicates from a representative experiment. Statistical significance (P4 and P5) was determined using a two-tailed Student's t-test. **\**P < 0.01. *ns* not significant. Source data are provided as a Source Data file for Fig. 2b, c. **d** At 24 h.p.i. of HRV-C3, airway organoids were fixed and immune-labeled with an α-VP3 (green) and α-ACCTUB (red). Nuclei and actin filaments were counterstained with DAPI (blue) and Phalloidin-647 (white), respectively. The experiment was independently performed three times with similar results. Scale bar, 10 μm.

passages of HRV-C3 (Fig. 3b, Supplementary Fig. 4). We also tested HRV-C11+ (Fig. 3c) and HRV-C15+ (Fig. 3d) medium (the medium collected from organoids infected with an HRV-C11+ and HRV-C15+ clinical specimen respectively) in nasal organoids derived from another donor. The two subtypes of HRV-C were serially passaged three times with high viral loads, and more passages could be furthered, if necessary. Overall, nasal organoids per se sustained serial propagation of HRV-C without any intervention. To verify the infectiousness, we demonstrated that HRV-C passaged multiple times in the nasal organoids established a productive infection in nasal organoids (Fig. 3e, Supplementary Fig. 5). HRV-C virions released from the nasal organoids were concentrated and applied to transmission electron microscope (TEM) examination. Virion particles sized around 30 nm, and empty capsid particles were readily discernible (Fig. 3f). We sequenced the whole genome of the viruses in the initial clinical specimen and the viruses after 1 and 6 consecutive passages with CYT treatment; no adaptive mutation was identified (Supplementary Information File). Collectively, airway and nasal organoids enabled reproducible virus isolation and serial passage of poorly cultivable HRV-C.

We moved on to examine the effect of temperature on the replication of HRV-C and HRV-A, as most rhinoviruses exhibit optimal growth at 33 °C rather than the core body temperature of 37 °C[30,31]. To this end, nasal organoids were inoculated with HRV-C3 and HRV-A1 at 33 °C or 37 °C. After washing to remove the residual inoculum, we continued to incubate the infected organoids at 33 °C or 37 °C and then harvested culture media to examine viral growth. The results showed that HRV-C grew significantly better at 37 °C than at 33 °C. We repeated the experiment in nasal organoids derived from two other donors and obtained similar results (Fig. 3g). In contrast, HRV-A replicated more robustly at 33 °C than at 37 °C (Fig. 3h), consistent with the growth advantage of most rhinoviruses at 33 °C as reported previously[1,30,31]. The results suggested that the optimal temperature of 33 °C for most rhinoviruses may not apply to HRV-C3.

## Nasal organoids more permissive to HRV-C than airway organoids

Given that airway organoids, but not nasal organoids, required CYT387-mediated immunosuppression to sustain serial virus passage, we postulated that nasal organoids might be more permissive to HRV-C than airway organoids, and/or the antiviral response may be more robust in airway organoids than nasal organoids, which restricted serial virus passage in the airway organoids. To test the first hypothesis, we inoculated nasal and airway organoids in parallel with serially diluted HRV-C inocula, i.e. from 1000 - 1 viral gene copy/cell, and monitored viral growth. With higher inoculations at 1000 copy/cell and 100 copy/cell, HRV-C replicated in airway organoids, albeit with a significantly lower viral load than in nasal organoids. After inoculation at 10 copy/cell and 1 copy/cell, we only observed viral growth in nasal organoids, whilst no viral growth was discernible in airway organoids (Fig. 4a). We obtained similar results in experiments independently performed in another pair of airway and nasal organoids from different donors (Fig. 4b). In the third pair of airway and nasal organoids, the virus replicated in airway organoids after a 10 copy/cell inoculation, yet the viral load was significantly lower than that in nasal organoids (Fig. 4c). Overall, the results indicated that nasal organoids were more susceptible to HRV-C and sustained more active viral replication than airway organoids.

We then examined the HRV-C infection rate and cellular tropism in nasal and airway organoids by flow cytometry analysis. The viral antigen VP3+ cells were more abundant in nasal organoids than airway organoids (Fig. 4d), which is consistent with the higher susceptibility of nasal organoids to the virus than airway organoids (shown in Fig. 4a, b). Most VP3+ cells in nasal and airway organoids were ACCTUB positive, indicating that HRV-C primarily infected ciliated cells, consistent with the prior findings in human primary airway epithelial cells[32]. Scan electron microscope (SEM) revealed damaged and deformed cilia in the infected airway organoids, in

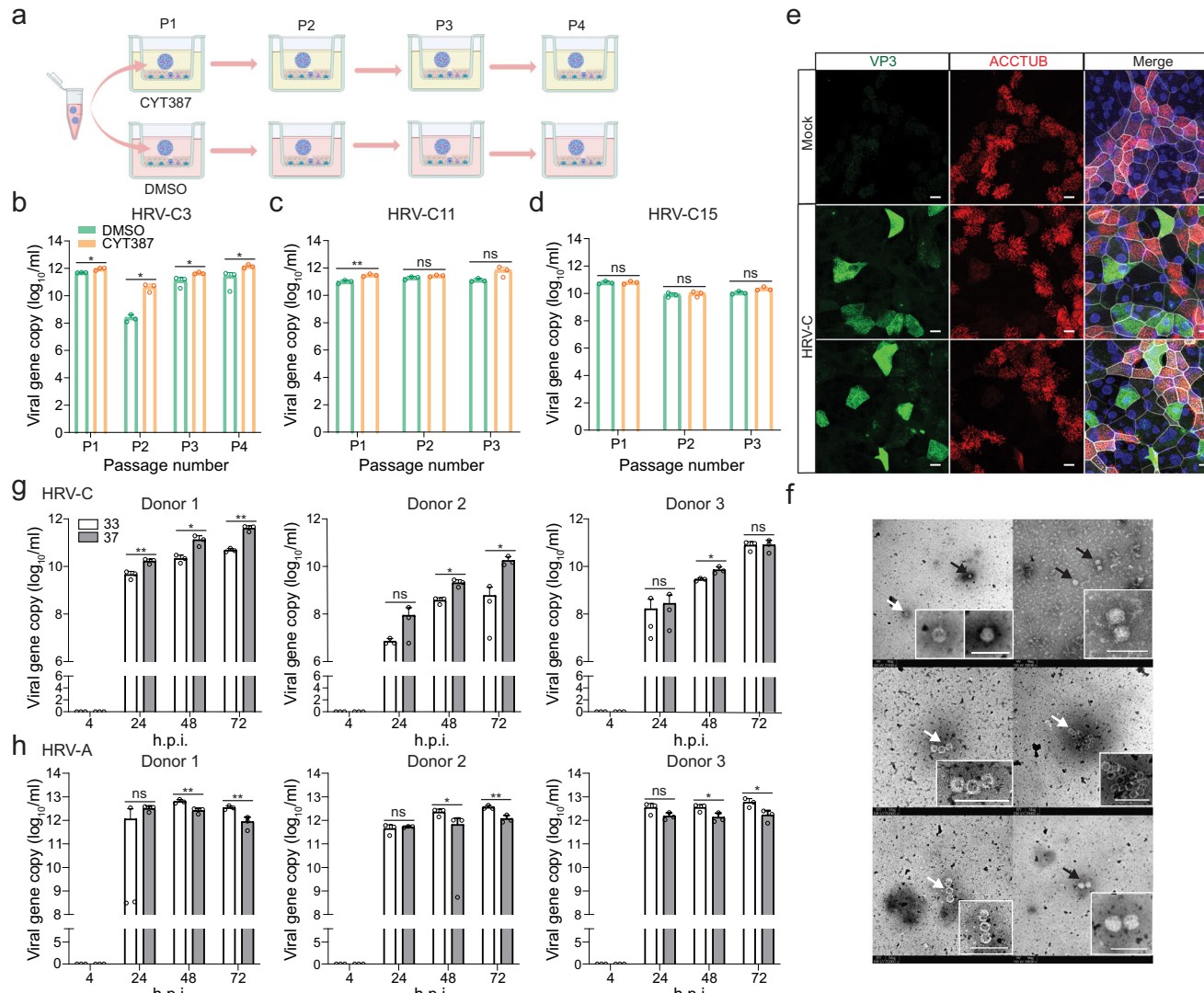

**Fig. 3 | Human nasal organoid sustained serial HRV-C propagation. a** A schematic graph outlines the experimental procedure for panels (**b**, **c**, and **d**) created with Biorender.com. **b–d** Nasal organoids pre-treated with CYT387 or DMSO were inoculated with HRV-C3 (**b**), C11 (**c**), and C15 (**d**) at 100 viral gene copy/cell and incubated in the presence of CYT387 or DMSO respectively (*n* = 3). CYT387- and DMSO-treated media were brought forward as inoculum to the next round of infection, during which CYT387 and DMSO treatment continued. Culture media were harvested at 96 h.p.i. to detect viral replication. **e** At 24 h.p.i., nasal organoids inoculated with HRV-C3 were fixed and doubled stained with an α-VP3 (green) and α-ACCTUB (red). Nuclei and actin filaments were counterstained with DAPI (blue) and Phalloidin-647 (white), respectively. Scale bar, 10 μm. **f** TEM images of HRV-C3 viral particles in the culture media of infected nasal organoids. Arrows indicate viral particles (black) or empty capsids (white). Scale bar, 100 nm. **g**, **h** Nasal organoids derived from 3 different donors inoculated with HRV-C3 (**g**) and HRV-A1 (**h**) were incubated at 33 °C or 37 °C (*n* = 3). The culture media were harvested at the indicated h.p.i. to detect viral replication. Data represent the mean and SD of the indicated number (n) of biological replicates from a representative experiment. Statistical significance (in **b**, **c**, **d**, **g**, and **f**) was determined using a two-tailed Student's t-test. \**P* < 0.05, \*\**P* < 0.01. *ns* not significant. The experiments in e and f were independently performed three times with similar results. Source data are provided as a Source Data file for Fig. 3b, c, d, g, and h.

contrast to the dense and delicate cilia in the mock-infected organoids (Fig. 4e).

## A more robust innate immune response in airway organoids than nasal organoids

We next asked whether the antiviral response in HRV-C-infected airway organoids was more robust than that in nasal organoids, thus restraining serial viral propagation in the airway organoids. We performed RNA sequencing to systematically characterize the cellular response in the airway and nasal organoids at 48 h post HRV-C infection and assess the effect of CYT treatment during the infection. Compared to the mock-infected organoids, we observed a notable upregulation of inflammatory cytokine CXCL10 (IP-10) and interferon-stimulated genes (ISGs), including OASL, IFIT1, ISG15, and IFI44L in the

airway organoids (Fig. 5a). CXCL9, ZBP1, and RSAD2 were markedly induced as well. The most upregulated genes in the virus-infected airway organoids were also highly induced in the infected nasal organoids. Yet, HRV-C infection triggered a more extensive and intensive cellular response in airway organoids than in nasal organoids. A total of 299 and 47 genes were significantly upregulated in airway organoids and nasal organoids, respectively; nasal organoids shared 34 out of the 47 upregulated genes with airway organoids (Fig. 5b). CYT treatment markedly downregulated the highly induced genes by HRV-C infection, with a more prominent effect in airway organoids than in nasal organoids (Supplementary Fig. 6a). A total of 443 and 170 genes were significantly downregulated in CYT-treated HRV-C infected airway and nasal organoids, respectively, compared to mock-treated counterparts. Nasal organoids shared 80 among 170 downregulated genes

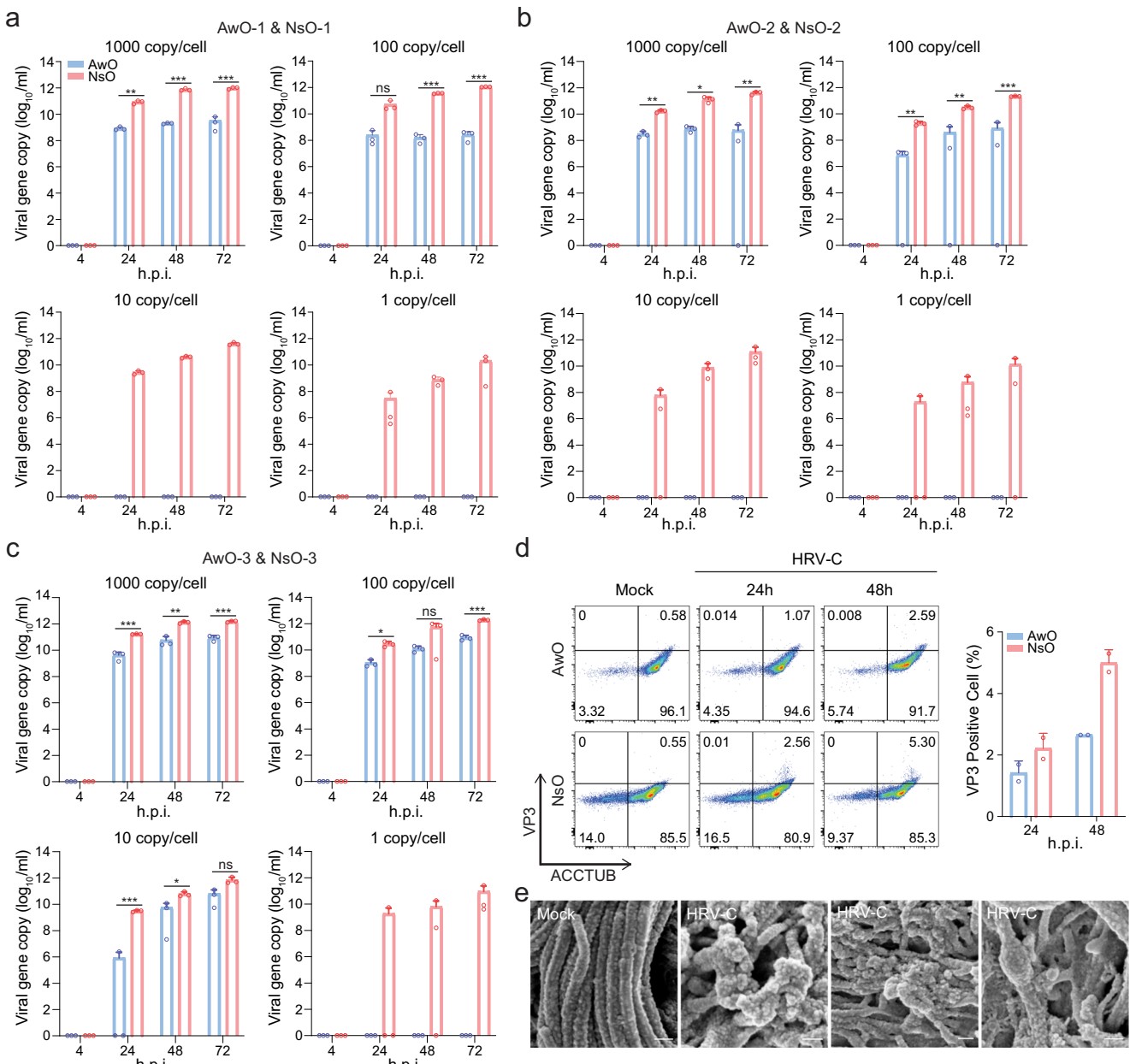

**Fig. 4 | Nasal organoids were more permissive to HRV-C than airway organoids.**
**a**–**c** Airway (AwO) and nasal (NsO) organoids derived from different donors were inoculated in parallel with HRV-C3 at 1000, 100, 10, and 1 viral gene copy/cell (*n* = 3). Culture media were harvested from the infected organoids at the indicated h.p.i. to detect viral replication. **d** After co-staining with α-VP3 and α-ACCTUB, HRV-C3- and mock-infected airway and nasal organoids were applied to flow cytometry analysis (*n* = 2). Representative histograms are shown on the left. Data on the right represent the mean and SD from a representative experiment. **e** SEM images of HRV-C3- and mock-infected airway organoids. The experiment was independently performed three times with similar results. Scale bar, 200 nm. Data represent mean and SD of the indicated number (n) of biological replicates from a representative experiment. Statistical significance (in **a**, **b**, and **c**) was determined using a two-tailed Student's t-test. *$P < 0.05$, **$P < 0.01$, ***$P < 0.001$. *ns* not significant. Source data are provided as a Source Data file for Fig. 4a–d.

with airway organoids (Supplementary Fig. 6b). Generally, CYT treatment blunted HRV-C-induced innate immunity more remarkably in airway organoids than in nasal organoids.

The differentially-expressed genes (DEGs) in the infected organoids versus the mock-infected counterparts belonged to GO terms such as "response to virus", "innate immune response", "cytokine/chemokine induction and signaling", etc. These GO terms were more notably enriched in HRV-C-infected airway organoids than in nasal organoids. Consistently, CTY treatment blunted the enriched GO terms more extensively in the infected airway organoids than in the nasal counterpart (Fig. 5c). Enriched pathways in HRV-C infected organoids involved IFN and inflammatory response, cell death, and

cilium biology (Fig. 5d). CYT treatment globally blunted the innate immunity in the infected nasal organoids, and more notably in airway organoids. Interestingly, cilium organization and movement were enhanced in the CYT-treated airway organoids. A more prominent induction of innate immunity in airway organoids than that in nasal organoids was verified in RT-qPCR assay (Fig. 5e). The transcriptional level of ISGs such as RIG-I, OASL and inflammatory cytokines IP-10, IL-6, and IL-8 was significantly higher in airway organoids than in nasal organoids at 24 and 48 h.p.i. We also analyzed viral reads in the RNA sequencing dataset. Consistent with the findings shown in Fig. 4, viral reads were significantly higher in nasal organoids than in airway organoids, and CYT-mediated virus enhancement was more

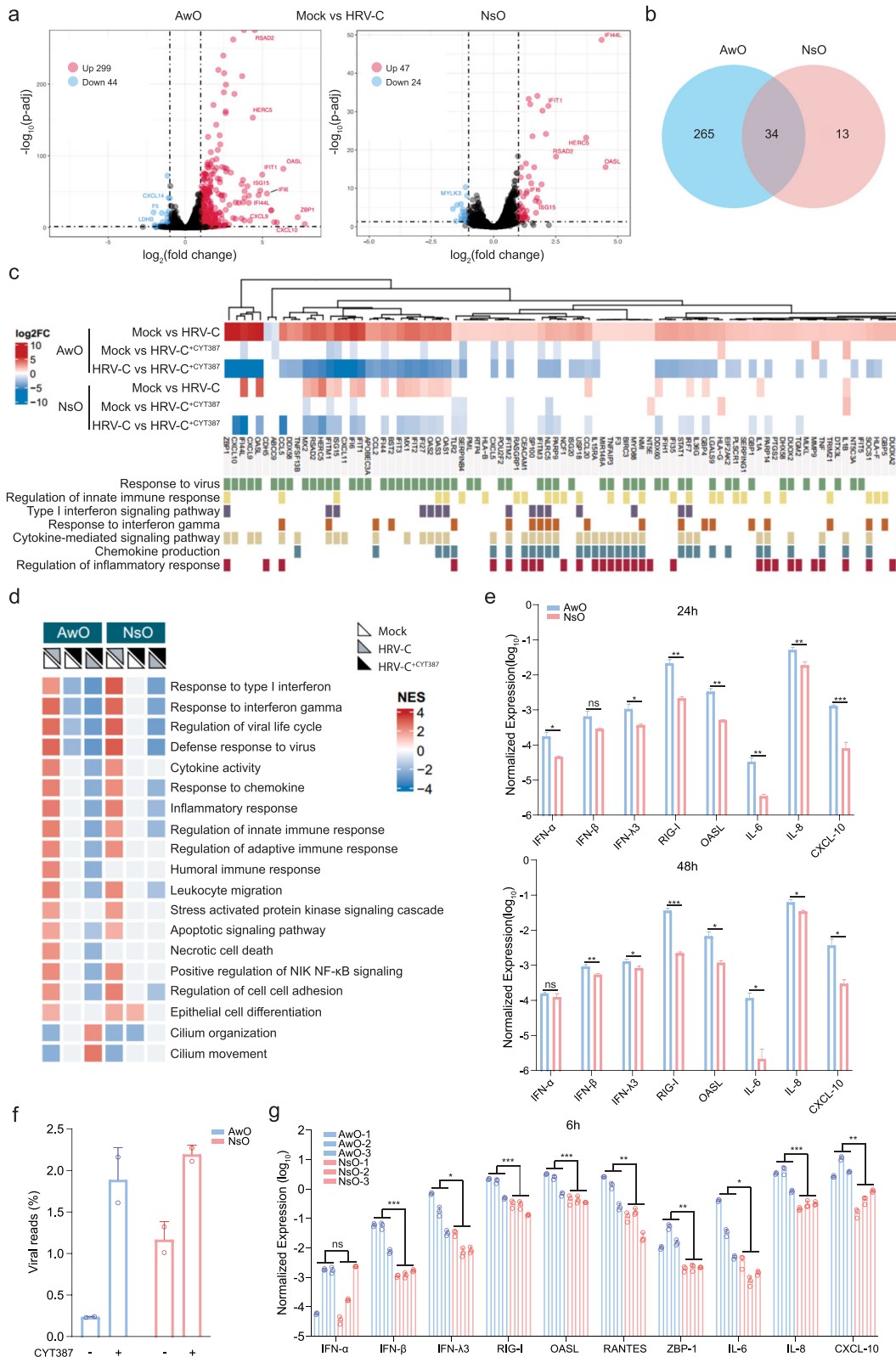

prominent in airway organoids than in nasal organoids (Fig. 5f). Collectively, these results revealed a stronger innate immune response in airway organoids than in nasal organoids, and elucidated the biological mechanism leading to the differential dependency of airway and nasal organoids on CYT for serial viral passage.

While nasal organoids were more susceptible to HRV-C than airway organoids and sustained more active viral replication, the infected airway organoids mounted a more robust innate immune response than nasal organoids. We asked whether the more prominent induction of innate immunity in airway organoids than nasal organoids was an attribute intrinsic to these organoids, or a particular manifestation in the context of virus infection. As aforementioned, nasal and airway organoids have a similar cellular composition, yet retain specific attributes of the upper and lower airway epithelium, respectively[33].

**Fig. 5 | Host transcriptional response in human airway and nasal organoids.**
HRV-C3 infected airway organoids (AwO) and nasal organoids (NsO) treated with
CYT387 (HRV-C$^{+CYT387}$) or DMSO (HRV-C), together with mock-infected organoids
(Mock) were applied to RNA sequencing analysis. **a** Volcano plot shows DEGs in the
infected airway and nasal organoids compared with mock-infected organoids.
DEGs with a log2(fold change) > 1 and < −1 are shown in red and blue, respectively.
**b** Venn diagram shows the numbers of unique and common upregulated DEGs in
the infected airway and nasal organoids. **c** Heatmap depicts DEGs in the indicated
organoids and the assigned GO biological processes (GO:0009615, GO:0045088,
GO:0060337, GO:0034341, GO:0019221, GO:0032602, GO:0050727). **d** The heat-
map demonstrates the enriched GO terms in the indicated airway and nasal orga-
noids. The color of the dots represents the normalized enrichment score (NES)

value for each enriched GO term. **e** GAPDH normalized expression levels of innate
immune molecules in HRV-C3-infected airway and nasal organoids at 24 and
48 h.p.i ($n = 3$). **f** Data show the percentage of virus-aligned reads over total reads in
the indicated organoids ($n = 2$). **g** Three lines of airway and nasal organoids were
treated with 10 μg/ml Poly(I:C) or mock-treated ($n = 3$). At 6 h post-treatment,
organoids were harvested for RT-qPCR. Results show the GAPDH normalized
expression level of innate immune molecules in the airway and nasal organoids.
Data represent mean and SD of the indicated number (n) of biological replicates
from a representative experiment. Statistical significance (in **e**, **g**) was determined
using a two-tailed Student's t-test. *$P < 0.05$, **$P < 0.01$, ***$P < 0.001$. ns not sig-
nificant. Source data are provided as a Source Data file for Fig. 5e, g.

This aligns with prior reports that the adult stem cell-derived human
intestinal organoids maintain architectural and functional character-
istics of intestinal segments from which they are derived since the
attributes of organoids were imprinted in the adult stem cells
accommodated in the original tissues[34]. To address whether innate
immunity is naturally more active in airway organoids than nasal
organoids, we performed a Poly(I:C) stimulation assay and examined
the immune induction, which would eliminate the potential con-
founding effect of differential viral replication in these two types of
organoids.

Poly(I:C), a synthetic virus mimic, induces innate immunity in
human primary cells and organoids[27]. We performed a Poly(I:C) sti-
mulation assay in 3 lines of airway organoids and 3 lines of nasal
organoids obtained from 6 different donors. A significantly higher
expression of type III IFNs, ISGs (RIG-I, OASL, ZBP-1), and proin-
flammatory cytokines (IL-6, IL-8, and IP-10) was observed 6 h after
Poly(I:C) stimulation in airway organoids (Fig. 5g). The higher innate
immune response in airway organoids remained at 10 h post-
stimulation (Supplementary Fig. 7c). Thus, the higher responsiveness
of airway organoids to viral infection seemed to be an intrinsic attri-
bute since Poly(I:C) invariably triggered a more intensive antiviral and
inflammatory response in airway organoids than in nasal organoids.

### Receptor blocking, antiviral inhibition, and virus titration in airway and nasal organoids

HRV infection is initiated upon virus binding to cognate receptors and
receptor-mediated endocytosis. CDHR3, the cellular receptor of HRV-
C[6], a member of the cadherin family of transmembrane proteins, is
highly expressed in human airway epithelium, especially in ciliated
cells in the native airway epithelium[35]. First, we examined CDHR3
expression and distribution in the airway and nasal organoids in
comparison with the undifferentiated counterparts that serve to
maintain and expand the organoid culture. Flow cytometry analysis
showed that the percentage of CDHR3+ cells increased from less than
20% in the undifferentiated organoids to over 60% and 80% in the
differentiated airway and nasal organoids, respectively (Fig. 6a). Con-
focal imaging revealed high CDHR3 expression in the differentiated
nasal and airway organoids (Fig. 6b, Supplementary Fig. 7); most
ACCTUB+ positive ciliated cells expressed CDHR3 on the apical sur-
face, a small portion of CDHR3+ cells were non-ciliated cells.

We then performed an antibody-blocking experiment in nasal
organoids to assess the role of CDHR3 on HRV-C replication. We
examined HRV-C replication in the nasal organoids in the presence of
two α-CDHR3 antibodies or an isotypic IgG. Blocking CDHR3 with
specific antibodies significantly decreased viral growth, verifying that
CDHR3 is necessary for HRV-C entry into nasal organoids (Fig. 6c). To
demonstrate the application of respiratory organoids for identifying
antiviral agents against HRV-C, we tested an HRV inhibitor, Rupintrivir
targeting HRV 3 C protease as a proof-of-concept. Itraconazole[32], a
repurposed antifungal drug with documented efficacy against rhino-
viruses, was also tested. We examined the effect of these two drugs
against HRV-C and HRV-A in airway organoids and observed a more

prominent inhibitory effect of both drugs on HRV-C than HRV-
A (Fig. 6d).

Currently, few in vitro assays are available to quantify HRV-C
infectious particles since no standard cell lines are susceptible to the
virus, except for an adaptive strain C15a which developed a cytopathic
effect and formed plaques in a CDHR3-expressing stable cell line
developed by Bochkov et al.[36]. The quantification of HRV-C primarily
relies on RT-qPCR assays to determine the viral gene copy number as
we demonstrated above. Given the susceptibility of airway and nasal
organoids to HRV-C, we sought to establish an organoid-based
immunofluorescence assay (IFA) to quantify HRV-C infectious vir-
ions. An aliquot of HRV-C collected from the infected organoids with
$1.2 \times 10^{11}$ viral gene copy/ml was applied to inoculate nasal organoids
in triplicate after serial dilution. The organoid monolayers were fixed at
24 h.p.i. and applied to immunostaining with an α-VP3 antibody to
label HRV-C infected cells, followed by analysis in a high-content
imaging analysis system. Figure 6e shows representative images of the
organoids infected with serially diluted virus. A total of 3 areas were
randomly selected from each organoid monolayer, from which the
number of VP3+ cells was calculated. The average number of VP3+ cells
was shown in Fig. 6f. We calculated the virus titer based on the image
analysis of organoids inoculated with the highest dilution factor $10^3$;
the number of infectious particles in the sample was around $3.7 \times 10^6$
IFU/ml.

## Discussion

We have established the first organoid culture system of human
respiratory epithelium from primary lung tissues and nasal epithelial
cells that allows for the reconstitution and expansion of the entire
human respiratory epithelium in culture plates with excellent effi-
ciency and stability[13–17]. Given that nasal and airway organoids accu-
rately simulate the native airway epithelium, we hypothesized that
these respiratory organoids may sustain a productive infection of HRV-
C, a common respiratory virus refractory to routine virus cultivation.
Moreover, the high stability and easy availability of these respiratory
organoids offered by the robust culture system may provide a repro-
ducible cultivation system for HRV-C, eliminating a long-standing
obstacle to understanding this common human respiratory virus.
Indeed, after inoculation of HRV-C+ clinical specimens, we observed an
active viral growth, indicating that airway organoids can recapitulate
the susceptibility of the human airway epithelium to HRV-C (Fig. 1b).
However, serial HRV-C propagation in airway organoids was unsuc-
cessful unless an immuno-suppressive molecule CYT387 was applied
to attenuate the antiviral response triggered in the infected organoids
(Fig. 2b, c).

After the first demonstration of HRV-C infection in surgically
resected sinus mucosa and nasal polyps[4], Bochkov et al. identified the
cellular receptor CDHR3 for HRV-C[6]. The landmark discovery was fol-
lowed by the establishment of a stable Hela-E8 cell line carrying a
higher cell-surface CDHR3 expression variant. Through serial passage
in the Hela-E8 CDHR3 cells, one HRV-C15a strain with enhanced fitness
was obtained due to adaptive mutations[36]. However, the two adaptive

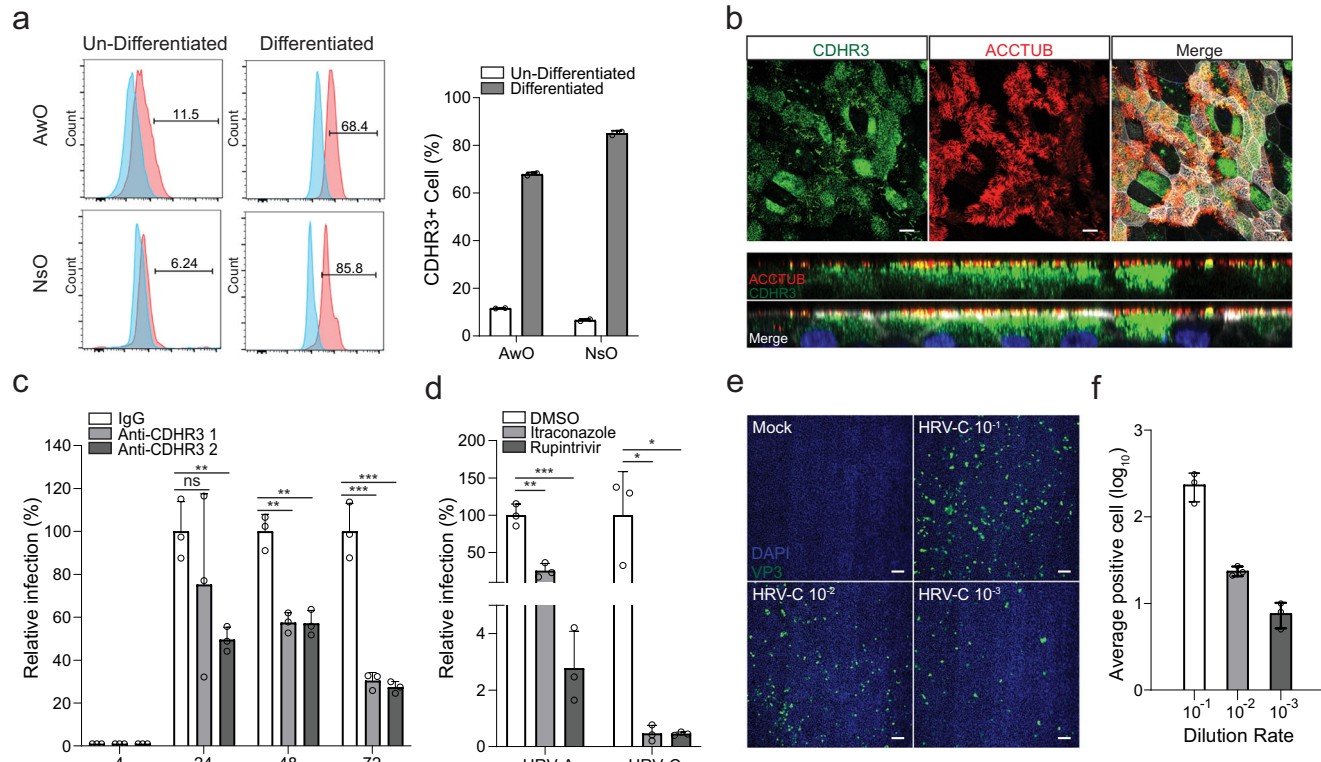

**Fig. 6 | Receptor blocking, antiviral inhibition, and virus titration in airway and nasal organoids. a** Airway (AwO) and nasal organoids (NsO) and their un-differentiated counterparts were immune-stained with an α-CDHR3 and an isotype IgG control and applied to flow cytometry analysis (n = 2). The left panel shows the representative histograms. **b** Nasal organoids were fixed and double-stained with an α-CDHR3 (green), α-ACCTUB (red). Confocal images of en face (top) and cross-section (bottom) are shown. Nuclei and actin filaments were counterstained with DAPI (blue) and Phalloidin-647 (white), respectively. Scale bar, 10 μm. **c** Nasal organoids were treated with two α-CDHR3 of 20 μg/ml and autologous IgG for 2 h (n = 3), followed by HRV-C3 inoculation and further incubation with the same antibodies. Culture media were harvested at the indicated h.p.i. to detect viral replication. Results show the relative viral load in the α-CDHR3 treated organoids versus those in IgG-treated organoids. **d** Airway organoids infected with HRV-A1 or HRV-C3 were treated with 1 μM Rupintrivir or 3 μM Itraconazole or DMSO (n = 3). Culture media were harvested from the infected organoids at the indicated h.p.i.

and applied to the detection of viral replication by RT-qPCR. Results show the relative viral load in the drug-treated organoids versus those in DMSO-treated organoids. **e, f** Nasal organoids were infected with HRV-C3 after 10-fold serial dilutions ($10^{-1}$ - $10^{-3}$). At 24 h.p.i., the organoid monolayers were fixed and immune-stained with an α-VP3, followed by imaging analysis. **e** Representative high-content confocal images show the VP3+ cells (green) in the monolayers infected by serially-diluted viruses. Nuclei were stained with DAPI (blue). Scale bar, 100 μm. **f** Three different areas in each organoid monolayer were randomly selected for calculating VP3+ cells. The results show the average number of VP3+ cells at different dilutions. Data represent mean and SD of the indicated number (n) of biological replicates from a representative experiment. Statistical significance (in **c, d**) was determined using a two-tailed Student's t-test. *$P < 0.05$, **$P < 0.01$, ***$P < 0.001$. ns not significant. The experiments in **b** and **e** were performed three times independently with similar results. Source data are provided as a Source Data file for Fig. 6a, c, d, and f.

mutations only partially conferred the enhancing phenotype to other recombinant HRV-C subtypes. Thus far, this C15a appeared to be the only strain that can be propagated to a yield adequate for experimentations. Moreover, the adaptive mutations were cell-type specific; HRV-C15a virus propagated in Hela-E8 CDHR3 cells appeared to replicate less actively than wildtype C15 virus in human primary bronchial epithelial cells[36], which may hinder applications of the adaptive virus strain for understanding human HRV-C respiratory infections. In this study, we randomly selected three HRV-C+ clinical specimens with different subtypes, all of which were serially passaged in nasal organoids with excellent efficiency and notable viral genome stability (Fig. 3b, c,d, Supplementary Information File). Thus, we have established a robust cultivation system to reproductively propagate the poorly cultivable HRV-C.

The unique advantage of human adult stem cell-derived intestinal and respiratory organoids for isolating and propagating uncultivable viruses lies in their high biological relevance to the native intestinal and respiratory epithelium, respectively[37], the primary target of many viruses. These epithelial organoids are equipped with most, if not all, host components to sustain the infection and propagation of human viruses, including viruses uncultivable in standard cell lines. Primary

airway/bronchial epithelial cells, which are commercially available, may have the essential host proteins that render them susceptible to human respiratory viruses. Yet the limited expandability of these primary cells, including other primary epithelial cell cultures, constrained their application for isolating and passaging viruses. Norovirus, a leading cause of viral gastroenteritis in humans and animals, has highlighted the unrivaled strength of epithelial organoids for propagating and studying uncultivable viruses. A long-awaited cultivation system seemed to be within immediate reach with the introduction of human adult stem cell-derived intestinal organoids[38]. However, while multiple norovirus strains grew and replicated in differentiated intestinal organoids[39], only a single strain GII.4 Sydney could be limitedly passaged in intestinal organoids[40,41]. Norovirus infection triggered a robust antiviral response in intestinal organoids, which prevented serial virus passage. However, individual depletion of major components of antiviral signaling pathways such as MAVS, STAT1, and STAT2 failed to achieve sustained virus propagation[41].

In this study, a simple CYT387 treatment enabled consecutive HRV-C passage in airway organoids, yet nasal organoids themselves reproducibly sustained serial virus passage in the absence of CYT387. The distinct dependency on CYT387 for sustained virus cultivation

prompted us to examine HRV-C replication fitness and virus-induced cellular response in these two types of organoids since reproducible virus cultivation is the outcome of an arms race between virus replication capacity and host antiviral immunity. We found that nasal organoids were indeed more susceptible to HRV-C than airway organoids and sustained more robust viral replication (Figs. 3, 4). Moreover, airway organoids mounted a more robust antiviral response than nasal organoids as revealed in HRV-C infection and verified in the Poly(I:C) stimulation assay (Fig. 5, Supplementary Fig. 6c). Thus, CYT387-mediated immunosuppression was essential for airway organoids to effectively dampen the vigorous antiviral response, boosting viral replication and enabling serial passage.

Mihylova et al. have reported that primary nasal epithelial cells mount a more robust antiviral response than bronchial epithelial cells upon rhinovirus B infection or RIG-I stimulation[42]. Nonetheless, our studies, including a comprehensive RNA sequencing analysis, individual profiling in virus infection, and Poly(I:C) stimulation, consistently demonstrated a higher level of immune activation in airway organoids than in nasal organoids. In our prior studies, we had a similar finding that SARS-CoV-2 infection triggered innate immunity more intensively in airway organoids than in nasal organoids[26], although the virus replicated less actively in the former than in the latter. More investigations, e.g., in paired nasal and airway organoids from the same patients, are warranted to clarify the discrepant findings between ours and Mihylova et al. We also demonstrated an optimal growth of HRV-A at 33 °C rather than 37 °C (Fig. 3h), in line with prior reports of most rhinoviruses[31]. However, we found that HRV-C3 exhibited a temperature preference of 37 °C rather than 33 °C (Fig. 3g). Ashraf et al. reported a distinct temperature preference of HRV-C from HRV-A and HRV-B in human primary sinus epithelial cells, in which the replication of HRV-C15 and C41 was comparable at 34 °C and 37 °C[5]. Here, further investigations by testing more HRV-C subtypes in the respiratory organoids are warranted to verify the temperature preference of HRV-C.

We observed abundant expression of HRV-C receptor CDHR3 on the apical surface of the airway and nasal organoids, particularly on the apical surface of ciliated cells (Fig. 6a, b, Supplementary Fig. 7), consistent with the result of a single-cell RNA sequencing study of native human respiratory epithelial cells[35]. Blocking CDHR3 with specific antibodies significantly suppressed HRV-C viral growth (Fig. 6c). We also demonstrated the effectiveness of two antiviral drugs against HRV as a proof-of-concept (Fig. 6d). Furthermore, we developed an organoid-based IFA to quantify HRV-C infectious particles (Fig. 6e, f). Collectively, our studies showcase the unique strength of respiratory organoids for studying HRV-C, from reproducible propagation of the poorly cultivable virus to detailed dissection of virus-host interaction and developing antivirals. Given that receptor blockage with α-CDHR3 suppressed viral growth, an organoid-based neutralization assay can be developed to evaluate the neutralizing activity of vaccination sera and engineered antibodies against HRV-C. More importantly, our study establishes a new paradigm for propagating and studying other uncultivable human and animal viruses.

## Methods

### Establishment, maintenance, and differentiation of respiratory organoids

This study was approved by the Institutional Review Board of the University of Hong Kong/Hospital Authority Hong Kong West Cluster (UW13-364 and UW21-695). Informed consent was obtained from patients and volunteers to collect human lung tissue and nasal cells. Multiple lines of organoids were established from surgically resected human lung tissues according to our previously published protocols[13,15,16,18]. To derive lung organoids, we used normal lung tissues adjacent to the diseased tissues, which were provided to us randomly. These lung tissues typically contained bronchioles of varying

sizes, surrounded by alveolar sacs. After 1 to 2 passages, the fibroblasts and other non-epithelial elements in the initial culture gradually diminished. Subsequently, the culture of pure epithelial organoids was stably expanded in the expansion medium (BiomOrgan Ltd) for over one year. Nasal organoids were derived from nasal epithelial cells noninvasively collected from healthy donors with perfect efficiency and consecutively passaged in the expansion medium for up to 6 months[17]. The undifferentiated lung and nasal organoids were passaged every 2 to 3 weeks with a ratio between 1:3 to 1:10, depending on whether mechanical shearing or enzymatic digestion was used to split the organoids. Proximal differentiation protocols for generating mature airway and nasal organoids were described previously[16,17]. Mature airway and nasal organoid monolayers growing on transwell inserts were used throughout this study unless indicated otherwise.

### Virus isolation, infection, and detection

A total of 9 HRV-C positive archived nasopharyngeal specimens from clinical patients were used for the study. Mature human airway and nasal organoids growing on 24-transwell inserts were washed twice with the basal medium (Advanced DMEM/F-12 (Gibco) supplemented with 1% HEPES, 1% GlutaMAX, and 1% Penicillin/Streptomycin), followed by virus inoculation with the indicated viral gene copy and subsequent incubation in the PD medium (PneumaCult-ALI medium (STEMCELL Technologies) supplemented with 10 µM DAPT and 10 µM Y27632), at 37 °C for 4 h. After incubation, the airway or nasal organoids were washed 3 times with the basal medium to remove residual inoculum. The apical and basolateral chambers were replenished with 300 µl and 500 µl PD medium, respectively. As for CYT387 treatment, airway and nasal organoids were pre-incubated in the PD medium with 1 µg/ml CYT387 (Invivogen, inh-cy87) or DMSO overnight. After virus inoculation, PD medium supplemented with 1 µg/ml CYT387 or DMSO was dispersed into apical and basolateral chambers to maintain the organoids. To assess the effect of temperature on viral growth, we infected nasal organoids with HRV-C3 and HRV-A1 at 100 viral gene copy/cell and maintained the organoids at 33 °C or 37 °C. To demonstrate the effect of CDHR3 for HRV-C viral replication, we pretreated organoids with two antibodies against CDHR3 (Abcam, ab121337; Sigma-Aldrich, HPA011218) of 20 µg/ml and autologous IgG (Abcam, ab172730) for 2 h, followed by HRV-C3 inoculation at 1000 viral gene copy/cell and further incubation in the presence of α-CDHR3 and IgG respectively. Culture media were harvested from the infected organoids at the indicated h.p.i. to detect viral replication. To investigate the antiviral effect of inhibitors on HRV-C replication, organoids were inoculated with HRV-A1 or HRV-C3 at 1000 viral gene copy/cell and incubated with the PD medium in the presence of 1 µM Rupintrivir, 3 µM Itraconazole, or DMSO. The culture media were collected at the indicated h.p.i. for viral load detection.

To examine viral replication, we harvested cell-free culture media from top chambers at the indicated hours post-infection, followed by RNA extraction using the RNAeasy Mini Kit (Qiagen) and detection of viral loads with the RT-qPCR assay targeting human rhinovirus 5′UTR gene using a QuantiNova Probe RT-PCR Kit (Qiagen) and primers listed in Supplementary Table 1. Thermal cycling conditions included reverse transcription at 45 °C for 10 min, initial denaturation at 95 °C for 5 min, followed by 40 cycles of 95 °C for 5 s, and 55 °C for 30 s. All experiments with live viruses were conducted in biosafety level 2 laboratories.

### Immunofluorescence staining and confocal imaging

To identify the virus-infected cells, we stained the organoids and performed confocal imaging as described elsewhere[20,21,25,28]. Briefly, airway and nasal organoids were infected or mock-infected with HRV-C3 at 1000 viral gene copy/cell and incubated for 24 h. After fixation with 4% paraformaldehyde (PFA) at room temperature for 1 h, membrane inserts seeded with organoids were cut and removed from

transwells, and applied to immune staining. The fixed organoid monolayers were permeabilized in 0.1% Triton-X100 at room temperature for 10 min and blocked with 3% bovine serum albumin (BSA) for 1 h. Subsequently, organoids were incubated with primary antibodies, including α-Rhinovirus VP3 (Invitrogen, MA5-18249), α-CDHR3 (Abcam, ab121337), α-ACCTUB (Sigma-Aldrich, T7941; Abcam, ab179504) at 4 °C overnight, followed by secondary antibodies (Supplementary Table 1). Nuclei and actin filaments were counterstained with DAPI (ThermoFisher) and Phalloidin-647 (Sigma-Aldrich), respectively. The organoids were then whole mounted on a glass slide with ProLong™ G Sigma-Aldrich lass Antifade Mountant (Invitrogen). The confocal images were acquired using a Carl Zeiss LSM 980 confocal microscope.

### Flow cytometry analysis
Airway and nasal organoids were infected or mock-infected with HRV-C at 10000 viral gene copy/cell. At 24 and 48 h.p.i, the organoids were dissociated into single cells with 10 mM EDTA (Invitrogen) at 37 °C for 30–60 min and fixed with 4% PFA for 30 min at room temperature. After permeabilization with 0.1% Triton X-100 for 5 min at 4 °C, cells were incubated with primary antibodies, α-Rhinovirus VP3 (Invitrogen, MA5-18249), α-ACCTUB (Abcam, ab179504), α-CDHR3 (Abcam, ab121337) for 1 h, followed by secondary antibodies (Supplementary Table 1). The immune-stained cells were re-suspended in 2% FBS/PBS and analyzed using Agilent NovoCyte Quanteon Analyzer. FlowJo software was used for data processing.

### Scanning and transmission electron microscopy
Virus- and mock-infected airway organoids were fixed with 2.5% glutaraldehyde (GTA). HRV-C3 virions in culture media were concentrated by ultracentrifugation and fixed with 2.5% GTA. Sample processing was performed by the Electron Microscope Unit of the University of Hong Kong. The LEO 1530 FEG Scanning Electron Microscope and Philips CM100 Transmission Electron Microscope were used for image acquisition.

### RNA sequencing analysis
Airway and nasal organoids were pretreated with CYT387 or DMSO, and infected with HRV-C3 at 10000 viral gene copy/cell or mock-infected, followed by further incubation with CYT387 or DMSO for 48 h. HRV-C3-infected airway and nasal organoids with CYT387 or DMSO treatment, together with mock-infected organoids, were applied to RNA sequencing analysis. Samples were harvested for RNA extraction using RNAeasy Mini Kit (Qiagen). The cDNA libraries were prepared with KAPA mRNA HyperPrep Kit according to the manufacturer's protocol. Quality control of raw fastq data was carried out by FastQC v0.11.7 (http://www.bioinformatics.babraham.ac.uk/projects/fastqc/) and fastp. Clean reads were aligned to the UCSC GRCh38 reference and the HRV-C3 strain genome (GenBank: OK161378.1) using Hisat2 v2.2.1. HTSeq v0.6.1 was used to generate the raw read count for each gene. Differential expression analysis was performed using DESeq2. Genes with log2|fold change| > 1 and adjusted $p$-value < 0.05 were considered significantly different. The volcano plot was generated using the EnhancedVolcano R package (https://github.com/kevinblighe/EnhancedVolcano). Heatmaps of gene expression levels were constructed using the Pheatmap R package (https://cran.r-project.org/web/packages/pheatmap/index.html). Gene set enrichment analysis (GSEA) was performed using the fgsea R package (https://github.com/ctlab/fgsea). Analysis and visualization of RNA-seq data were carried out in R v4.3.2. Volcano plots of the DEGs were generated using ggplot2 v3.5.1 R package. Heatmaps of comparison and pathway were created by the ComplexHeatmap v2.18.0 R package. The RNA-Seq data were uploaded to NCBI Gene Expression Omnibus (GEO) under accession number GSE254400 (token ajavwwmqfbaptch).

### Quantification of cellular RNA transcript by RT-qPCR
Airway and nasal organoids were infected with HRV-C3 at 10000 viral gene copy/cell or mock-infected. Cell lysates were collected 24 and 48 h.p.i. For Poly(I:C) stimulation assay, airway and nasal organoids were treated or mock-treated with 10 μg/ml Poly(I:C), and cell lysate was collected at 6 and 10 h post-stimulation. Subsequently, total RNAs were extracted from organoids with an RNeasy Mini kit (Q), and reverse transcribed into cDNA using PrimeScript™ RT Reagent Kit (Takara). qPCR was performed with TB Green Premix Ex Taq II (Takara) and specific primers (Supplementary Table 2) using a LightCycler® 96 Instrument (Roche) as described elsewhere[21,43–45].

### Organoid-based immunofluorescence assay (IFA)
The culture media harvested from HRV-C3 infected nasal organoids, after a 10-fold serial dilution, were used to inoculate nasal organoids. At 24 h.p.i., the organoid monolayers were fixed and permeabilized with 0.1% Triton X-100 for 10 min and blocked with 3% BSA for 1 h, followed by incubation with α-Rhinovirus VP3 (Invitrogen, MA5-18249), and a secondary antibody. Nuclei were counterstained with DAPI (Thermo Fisher). The organoids were then whole-mounted on a glass slide with ProLong™ Glass Antifade Mountant (Invitrogen). High-content confocal images were acquired using an IN Cell Analyzer 6500HS (GE Healthcare), a laser-based line scanning high-content imaging system. Image processing was performed using the IN Carta and Image J software.

### Statistical analysis
Statistical analysis was conducted using GraphPad Prism 9.0. Two-tailed Student's t-test was used to determine statistical significance. The number of replicates is indicated in figure legends. *$P < 0.05$, **$P < 0.01$, ***$P < 0.001$.

### Reporting summary
Further information on research design is available in the Nature Portfolio Reporting Summary linked to this article.

## Data availability
The RNA-seq data are available in the GEO database under accession code GSE254400. All further relevant source data supporting the findings of this study are available from the corresponding author upon written request. Source data are provided in this paper. Source data are provided with this paper.

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

## Acknowledgements

We thank the Center of PanorOmic Sciences and Electron Microscope Unit, Li Ka Shing Faculty of Medicine, University of Hong Kong, for assistance in confocal imaging flow cytometry and electron microscopy. We thank Jonathan Ip, Allen Chu, and Wing-Mui Chan for their help in sequencing HRV-C isolates. This work was partly supported by funding from Commissioned program for control of infectious diseases (CID-HKU1-3), Commissioned research on the novel coronavirus disease (COVID1903010-project 11) of the Food and Health Bureau of the HKSAR government to J.Z.; Health@InnoHK, Innovation and Technology Commission, of the HKSAR government to K.Y.Y.; General Research Fund (GRF, 17105420, 17113724), and Collaborative Research Fund (CRF, C7042-21G) of the Research Grants Council of HKSAR government to J.Z.

## Author contributions

J.Z. and H.C. designed and supervised the study. J.Z. and K.Y.Y. provided grant support. C.L., Yifei.Y., Z.W., M.C.C, J.H., S.Z., X.Z., Q.L., Y.D., Y.Z., and W.X. performed the experiments. J.Z., C.L., Yifei.Y., M.Y., J.-P.C., X.L.,

Yang.Y., L.H., and H.Chu analyzed the data. C.C.-Y.Y., K.K.-Y.W., J.F.-W.C., and K.Y.Y. provided the human tissues and clinical specimens. J.Z., C.L., Yifei.Y., H.Clevers, and K.Y.Y. wrote the manuscript.

## Competing interests

J.Z., K.Y.Y., H.Clevers, C.L., and M.C.C. are listed as inventors on the patent of airway organoids (Patent No: ZL 2019 8 0037552.0), and nasal organoids (US 63/358,795). J.Z. is the founder of BiomOrgan Ltd. All other authors declare no competing interests.

## Additional information

[1]Department of Microbiology, Li Ka Shing Faculty of Medicine, The University of Hong Kong, Pokfulam, Hong Kong, China. [2]Centre for Virology, Vaccinology and Therapeutics, Hong Kong Science and Technology Park, Hong Kong, China. [3]Department of Surgery, Li Ka Shing Faculty of Medicine, The University of Hong Kong, and Queen Mary Hospital, Hong Kong, China. [4]Clinical Stem Cell Research Center, Peking University Third Hospital, Beijing, China. [5]BiomOrgan Ltd, Hong Kong, China. [6]State Key Laboratory of Emerging Infectious Diseases, The University of Hong Kong, Hong Kong, China. [7]Carol Yu Centre for Infection, The University of Hong Kong, Pokfulam, Hong Kong, China. [8]Oncode Institute, Hubrecht Institute, Royal Netherlands Academy of Arts and Sciences (KNAW), and University Medical Center (UMC) Utrecht, Utrecht, the Netherlands. [9]Roche Pharmaceutical Research and Early Development, Basel, Switzerland. [10]These authors contributed equally: Cun Li, Yifei Yu. ✉e-mail: jiezhou@hku.hk

