## [Peer Review File · Nature Communications]

REVIEWER COMMENTS

Reviewer #1 (Remarks to the Author):

Peer Review for Manuscript NCOMMS-24-22873-T

Summary

The manuscript titled “Human respiratory organoids sustained reproducible propagation of human rhinovirus C and elucidation of virus-host interaction” focuses on the development of an organoid-based system to propagate human rhinovirus C (HRV-C). The system uses primary tissue-derived airway and nasal cell monolayers on transwells to study interactions between HRV-C and host cells. The authors demonstrate the nasal model is susceptible to HRV-C infection and can sustain virus propagation without intervention, while their airway model requires CYT387-mediated immunosuppression for serial virus passage. This research provides insights into virus-host interactions and has implications for developing antiviral strategies and studying other uncultivable viruses.

The article needs a rewrite to improve clarity; it’s convoluted in places. The study contributes new knowledge to the field by providing a new method (but not a new model per se) for studying HRV-C and potentially other uncultivable viruses.

Breakdown of the Paper Elements

Title

The title is clear, concise, and reflects the content of the article. Replace ‘respiratory’ with ‘nasal’ for greater clarity of findings.

Abstract

The abstract provides a succinct summary of the study— although claims of significant differences should be supported by stats (p-values and sample size).

Introduction

The introduction offers adequate background information and context.

Literature Review

The literature review is satisfactory and up to date.

Methods

The methods are not described in adequate detail; some key information is lacking. Where were cells obtained from? What are the specific donor characteristics? Clarify whether airway or lung—use

consistent terminology throughout. Where in the lung was the sample obtained from—bronchi, bronchiole, or alveolus? What is the catalog number of CYT387 (Invivogen)? This will be crucial for replication of the study.

Ethical Concerns

This study was approved by the Institutional Review Board of the University of Hong Kong/Hospital Authority Hong Kong West Cluster (UW13-364 and UW21-695). There is no mention of what these approvals cover—does it include acquisition of primary cells by brushing or curettage? Be specific—what was the inclusion criteria of donors, for example?

Data Analysis

The data analysis methodology is not always appropriate, with statistical methods incorrectly applied for some of the data and therefore misinterpreted. There is often an insufficient sample size to apply a Student's t-test, as this form of analysis requires the data to follow a normal distribution (which cannot be demonstrated with only two data points) and is based on degrees of freedom (which in this case would be sample size – 1 = 1), resulting in an unreliable p-value. The RNA sequencing and RT-qPCR assays are well-explained and justified. The authors state that only GraphPad Prism was used but some plots (i.e., RNAseq) were generated in other ways and this should be stated.

Results

The results are presented in appropriate sections and supported by data, tables, and figures. They show that the organoid system can propagate HRV-C and show an immune response using RNASeq. In some places, the results are not described clearly in the text and some clarification of the language used/rewriting is needed to improve this.

Figure 1a: The images describing the model need to be larger and clearer—these look exactly like ALI cultures. Why are these called organoids?

Lines 85-87: A phylogenetic tree is needed to show these are different strains. Some stats to back up the conclusion about HRV-C15 circulating more.

Figure 1b: What volume was on the apical and basolateral side? Could the difference be due to dilution factor?

Figure 2d: Some very nice immunofluorescence images. Some more explanation of what is shown would be helpful. It doesn't look like the ciliated cells are being targeted here either (as suggested from the later flow cytometry)—limited colocalization of VP and ACCTUB.

Lines 109-117: These are methods, not results.

Line 125: Significantly different from what? It's nonsignificant on the graph.

Line 134-5: Remove 'intervention'. And you do not show infectious virus in these plots—it is copy number, so you need to change this conclusion here.

Figure 3 and Line 146: Include stats in text. What is the p-value?

Figure 3b: Not showing which donor is which. Compare to Line 147.

Figure 3 in general: All viral copy number, not infectious virus! Where is this data? Could the high copy number be from defective viral genomes?

Figure 4: In comparison of nasal organoid titers and airway titers—were they from the same individual? Again, all copy number, no infectious virus.

Figure 4e: Have you got lower magnification images? Difficult to make out.

Line 176: No stats to back this up. What is the p-value?

Figure 5g: Is that following treatment with CYT387?

Figure 6: Not sure this aligns with their aims and probably needs further work to exclude other factors. Or move to supplementary as not specific to HRV but all viruses.

Figure 7a-b: Need isotype or secondary-only control for the staining as this could be non-specific—especially for 7b. Like Figure 6, not sure how this all fits here. Should be included as part of the model characterization earlier on. Seems out of place here.

Figure 7c-d: This is useful and shows the potential application of the nasal organoids, which does not have the immunosuppressive agent.

Discussion

The discussion provides an overview of the results. The implications and limitations of the study are addressed, including the potential for the organoid system to study other uncultivable viruses and the differences in immune responses between organoid types.

Conclusions

The conclusions align well with the stated objectives. The potential applications of the organoid system in antiviral strategy development are suggested.

References

The references appear properly cited and up to date.

Figures and Tables

The figures and tables are mostly clear and informative but lack appropriate statistical tests. Some figures could be made bigger (1a and 2a diagram) and supplemental data necessary for understanding HRV-C clinical strain differences would be beneficial in the first figure.

Strengths of the Manuscript

The manuscript provides a system for propagating HRV-C, overcoming a significant challenge in the field. Others have shown this 10+ years ago in differentiated cell culture models (<https://pubmed.ncbi.nlm.nih.gov/23035218/>, reference #8), which appears to be the same model used here. The uniqueness of this manuscript is the immunosuppression that enhances viral replication in airway cells (not nasal).

It offers a detailed comparison of immune responses between different types of respiratory organoids. The study has some well-described procedures and results but details are missing as described above. The implications for future research and antiviral development are justified.

Major Weaknesses

Definition of 'organoid' needs to be stated early on.

Clarity of 'airway' definition and consistent terminology used.

The study primarily focuses on HRV-C, and while it mentions the potential for studying other uncultivable viruses, it does not provide empirical evidence for this application.

What is the sequence of HRV? Has it undergone any mutation from replicating with/without immune suppression?

Low replicate values, especially for early model validation (n=1 or 2), and therefore inappropriate statistical analysis applied.

Ethics clarification.

Minor Textual Errors

Page 4, Line 16: "in vivo characterization." - Consider specifying the types of in vivo characterizations referred to.

Page 5, Line 38: "Moreover, virus growth exhibited significant variations" - Consider explaining the nature of these variations for clarity.

Noteworthy Results

The development of an organoid-based system to propagate human rhinovirus C (HRV-C) by CYT387-treated airway cell cultures or untreated nasal cell cultures.

Significance to the Field

The work is significant as it may provide a useful method for studying HRV-C and potentially other uncultivable viruses (though this claim isn't strongly backed), despite being similar to previously established models.

Comparison to Established Literature

The method resembles differentiated cell culture models used over a decade ago, but with the unique addition of immunosuppression enhancing viral replication in airway cells. Relevant reference: <https://pubmed.ncbi.nlm.nih.gov/23035218/>.

Support for Conclusions and Claims

Some conclusions and claims need additional evidence, particularly regarding the statistical analysis and infection results.

Flaws in Data Analysis and Interpretation

The data analysis has flaws, notably the inappropriate use of the Student's t-test with insufficient sample

sizes, resulting in unreliable p-values.

Methodology Soundness

The methodology has potential but lacks adequate detail in several areas, such as donor characteristics and precise cell source locations, which need clarification for reproducibility.

Reproducibility Detail

Insufficient detail is provided in the methods for full reproducibility, specifically regarding donor characteristics, cell sources, and CYT387 catalog number.

Reviewer #2 (Remarks to the Author):

General:

The authors have previously established the first human respiratory organoid culture system, which is a significant achievement and highly appreciated. Using this previously established system, the authors aimed to develop an organoid-based system to reproducibly propagate HRV-C and characterize virus-host interactions using respiratory organoids.

Major Comments:

1. The data indicate that HRV-C infects cells in both airway and nasal organoids. However, it is not clear whether the data fully support the goals stated by the authors. For instance, what advantages does the organoid-based system offer over the established Air-Liquid Interface (ALI) culture? The manuscript seems to underappreciate the precedent work using primary human bronchial epithelial cells in ALI culture, which has been instrumental in numerous pioneering studies, including the discovery and characterization of HRV-C infection and related transcriptional and epigenetic studies.

Please see the following articles: PMID: 35570279, PMID: 33188283, PMID: 28472984, PMID: 38843491.

2. Further validation and evidence are required for the conclusion that nasal organoids are more susceptible to HRV-C than airway organoids. Due to the very small sample size, the current data are insufficient to fully account for potential experimental or individual variability. Therefore, the generalization of this finding in comparison between the airway and nasal organoid is premature.

3. The manuscript speculates on the broader significance of the organoid system for uncultivable human and animal viruses without providing sufficient evidence. The same speculation could be made with ALI cultures of human BECs. Thus, additional data are required to support these claims and demonstrate why this system would be more appropriate than the ALI culture of HBECs.

Minor Comments:

1. The legend and schema for Figure 1 need improvement. It is unclear whether they correspond to airway organoids, nasal organoids, or both. To improve the readability and comprehension of the data presented, clarity in figure legends and schematics is required.

2. In both Fig 2d and 3e, ciliated cells are underneath the two cells marked by Phalloidin, which seems

strange. It appears that the images are presented by maximum intensity projection. Including orthogonal views would help to indicate the infection of HRV-C in ciliated cells.

Reviewer #3 (Remarks to the Author):

REVIEWER COMMENTS

Reviewer #1:

Summary

The manuscript titled “Human respiratory organoids sustained reproducible propagation of human rhinovirus C and elucidation of virus-host interaction” focuses on the development of an organoid-based system to propagate human rhinovirus C (HRV-C). The system uses primary tissue-derived airway and nasal cell monolayers on transwells to study interactions between HRV-C and host cells. The authors demonstrate the nasal model is susceptible to HRV-C infection and can sustain virus propagation without intervention, while their airway model requires CYT387-mediated immunosuppression for serial virus passage. This research provides insights into virus-host interactions and has implications for developing antiviral strategies and studying other uncultivable viruses.

The article needs a rewrite to improve clarity; it’s convoluted in places. The study contributes new knowledge to the field by providing a new method (but not a new model per se) for studying HRV-C and potentially other uncultivable viruses.

We appreciate the reviewer’s comment. We have revised the manuscript accordingly. We would like to highlight the significant contribution of our study to the field, which is not limited to introducing a novel method for studying HRV-C and other uncultivable viruses.

Our study would be a landmark progress for HRV-C research, fundamentally exceeding previous HRV-C studies in which primary tissues and primary epithelial cell models were used. While virus isolation using primary tissues brought about many variations and challenges (specified in the Introduction on page 3), primary epithelial cells showed limited expansion capacity and hardly sustained serial propagation of HRV-C. In contrast, we utilized adult stem cell (ASC) derived nasal and airway organoids recently established by our team. Our nasal and airway organoids can be stably and consecutively expanded over half a year to one year, which is critically essential for serial virus propagation. In our previous publications¹⁻⁴ (cited in the manuscript as well), we have extensively characterized our respiratory organoids and demonstrated that these organoids faithfully simulate the cellular composition, architecture, and functionality of the native human nasal and airway epithelium. Furthermore, we have reported that these respiratory organoids adequately recapitulate human respiratory virus infections, including influenza and SARS-CoV-2¹⁻⁴. This study showcased a novel application of our edge-cutting respiratory organoids, eliminating a long-existing challenge for HRV-C research, rather than just a new method. Overall, we have created a novel and robust cultivation system to propagate and investigate previously unculturable viruses, thanks to the uniqueness and high biological relevance of these respiratory organoids.

Breakdown of the Paper Elements

Title

The title is clear, concise, and reflects the content of the article. Replace 'respiratory' with 'nasal' for greater clarity of findings.

We appreciate the reviewer's comment. In this study, we demonstrated that nasal organoids sustained serial HRV-C passage, while airway organoids enabled reproducible HRV-C propagation with the aid of CYT387-mediated immunosuppression. Moreover, we demonstrated that HRV-C infection triggered a more robust antiviral response in airway organoids than in nasal organoids, which constrained serial viral propagation in the former. Given that a significant portion of our findings are based on studies in airway organoids, we believe "respiratory organoids" would summarize our study more accurately. To clarify, we have specified the definition of "respiratory organoids" in the revised manuscript on page 4.

Abstract

The abstract provides a succinct summary of the study— although claims of significant differences should be supported by stats (p -values and sample size).

Thanks for the reviewer's comment. We have amended the statistics analysis in the manuscript.

Introduction

The introduction offers adequate background information and context.

Literature Review

The literature review is satisfactory and up to date.

Methods

The methods are not described in adequate detail; some key information is lacking. Where were cells obtained from? What are the specific donor characteristics? Clarify whether airway or lung—use consistent terminology throughout. Where in the lung was the sample obtained from—bronchi, bronchiole, or alveolus? What is the catalog number of CYT387 (Invivogen)? This will be crucial for replication of the study.

We have published a series of research papers in prestigious journals and cited these papers in the manuscript. Given the comments, we have revised the methodology section and prepared a schematic illustration (Fig. 1a) of nasal and airway organoid culture accordingly. In brief, upon ethical approval, we randomly received lung tissues from patients undergoing surgical resections due to various disease conditions that would otherwise be put into medical waste. We have never intentionally requested, and will never request, any lung tissues to derive organoids. These lung tissues typically contained bronchioles of varying sizes, surrounded by alveolar sacs. Nonetheless, we can derive organoids with near-perfect efficiency using lung tissues without selection (of donors, tissue locations), indicating the robustness of the culture

system. Based on our long-term investigations, including this study, experiment results from organoids of different donors are very consistent.

We have provided the catalog number of CYT387 in the revised manuscript on page 16.

Ethical Concerns

This study was approved by the Institutional Review Board of the University of Hong Kong/Hospital Authority Hong Kong West Cluster (UW13-364 and UW21-695). There is no mention of what these approvals cover—does it include acquisition of primary cells by brushing or curettage? Be specific—what was the inclusion criteria of donors, for example?

The ethical approvals UW13-364 and UW21-695 have been obtained for using patients' tissues or nasal epithelial cells. We obtained small pieces of normal lung tissue adjacent to the diseased tissues from patients undergoing surgical resections. Nasal cells were harvested noninvasively from nasal turbinates of healthy donors using flocked swabs (like doing a COVID-19 RAT). The methodology for deriving organoids has been clearly described in our previous publications^{3,4}. We randomly procedure tissues and nasal cells to derive organoids; there are no inclusion or exclusion criteria.

Data Analysis

The data analysis methodology is not always appropriate, with statistical methods incorrectly applied for some of the data and therefore misinterpreted. There is often an insufficient sample size to apply a Student's t-test, as this form of analysis requires the data to follow a normal distribution (which cannot be demonstrated with only two data points) and is based on degrees of freedom (which in this case would be sample size – 1 = 1), resulting in an unreliable p-value. The RNA sequencing and RT-qPCR assays are well-explained and justified. The authors state that only GraphPad Prism was used but some plots (i.e., RNAseq) were generated in other ways and this should be stated.

We appreciate the reviewer's comments. We have complemented the data analysis for RNAseq and have revised the data analysis accordingly.

Results

The results are presented in appropriate sections and supported by data, tables, and figures. They show that the organoid system can propagate HRV-C and show an immune response using RNASeq. In some places, the results are not described clearly in the text and some clarification of the language used/rewriting is needed to improve this.

We have revised the related part in the manuscript accordingly.

Figure 1a: The images describing the model need to be larger and clearer—these look exactly like ALI cultures. Why are these called organoids?

Thanks for the reviewer's comment. We have mentioned “Air-liquid interface cultures of the primary human airway and nasal epithelial cells” in the original manuscript on page 3. The major limitation is “the limited expansion capacity inherent to primary epithelial cells substantially restricts their application for routine experimentations”. The major distinction between primary epithelial cell culture and our respiratory organoids is that we have expansion culture to maintain and expand the organoids, and differentiation protocols to generate large amounts of physiologically active respiratory epithelial cells (page 4). The 2D organoid monolayers used in the study were nasal and airway organoids induced maturation on transwell inserts. Although air-liquid interface transwell plates are used for primary cells and organoids, our organoid culture is fundamentally different from the primary cell culture.

We have prepared a new Fig. 1a, a schematic diagram of our airway and nasal organoid culture. The images of 2D organoid monolayer cultured in a transwell insert have been enlarged in Fig. 1b to provide more details.

Lines 85-87: A phylogenetic tree is needed to show these are different strains. Some stats to back up the conclusion about HRV-C15 circulating more.

Phylogenetic trees were created using the HRV-C 5' UTR and VP4/VP2 sequences (ED Fig. 1 and ED Fig. 2). Nasopharyngeal specimens of HRV-C positive were randomly collected. Out of the eight isolated strains, four were HRV-C15 subtype. Nonetheless, we have deleted the previous claim accordingly.

Figure 1b: What volume was on the apical and basolateral side? Could the difference be due to dilution factor?

Fig. 1b has been changed to Fig. 1c in the revised manuscript. The volume of the medium on the apical and basolateral sides were 300ul and 500ul, respectively. The viral gene copy in the apical medium was over ten times higher than that in the basolateral. Despite a 0.67-fold lower volume in the apical than in the basolateral chamber, it does not affect the overall conclusion presented in Fig. 1c.

Figure 2d: Some very nice immunofluorescence images. Some more explanation of what is shown would be helpful. It doesn't look like the ciliated cells are being targeted here either (as suggested from the later flow cytometry)—limited colocalization of VP and ACCTUB.

We appreciate the reviewer's comment. We have added an orthogonal view of the confocal image (ED Fig. 3) to explain the IF results in more detail. HRV-C primarily targets ciliated cells, as shown by flow cytometry (Fig. 4d), consistent with prior findings in human primary airway epithelial cells⁵. However, HRV-C infection damaged and depleted ACCTUB+ cilium in the airway organoids, especially when the infection progressed (Fig. 4e). Thus, some VP3+ ciliated cells were ACCTUB negative.

Lines 109-117: These are methods, not results.

We respectfully disagree with the reviewer. On page 6, we specified the rationale for using CYT387, which is the biological basis of our hypothesis, not methodology.

Line 125: Significantly different from what? It's nonsignificant on the graph.

We have revised the description on page 6 as suggested.

Line 134-5: Remove 'intervention'. And you do not show infectious virus in these plots—it is copy number, so you need to change this conclusion here.

We have revised the description on page 6 accordingly.

We would emphasize that no *in vitro* models are currently available to quantify HRV-C infectious particles since no standard cell lines are susceptible to the virus. The only exception is an adaptive strain, C15a, which formed plaques in a CDHR3-expressing stable cell line (HeLa-E8)⁶. However, we discussed the limitations in the manuscript on pages 11 and 12. As such, HRV-C replication primarily relies on RT-qPCR assays to detect the viral gene copy number. This is also the reason why we sought to establish an organoid-based immunofluorescence assay (IFA) to quantify HRV-C infectious virions (Fig. 6e and 6f).

We fully agree that viral gene copy cannot represent infectious virions. However, we demonstrated that, with the aid of CYT treatment, the media collected from the organoids in each passage productively infected a new batch of organoids in the next passage (Fig. 2c), suggesting infectious virions were present in the medium. Moreover, the serially passaged HRV-C infected airway organoid effectively without CYT treatment. Collectively, these results allowed us to conclude that viruses serially passaged in airway organoids were infectious.

Figure 3 and Line 146: Include stats in text. What is the p-value?

Student's t-test was used to calculate the p-values. The p-values from P1 to P4 are 0.0219, 0.0318, 0.0100, and 0.0168, respectively.

Figure 3b: Not showing which donor is which. Compare to Line 147.

The data presented in Fig. 3b and ED Fig. 4 were generated in organoids from two different donors. We have labeled donor 2 in ED Fig. 4.

Figure 3 in general: All viral copy number, not infectious virus! Where is this data? Could the high copy number be from defective viral genomes?

As aforementioned, no *in vitro* models are currently available to quantify HRV-C infectious particles, except the adapted C15a strain. Thus, an increasing viral gene copy number was

applied to show viral replication and propagation. Moreover, we have shown that HRV-C replicated efficiently in nasal organoids even after inoculation at 1 viral gene copy/cell (Fig. 4a, 4b and 4c), suggesting the virus particles released from organoids were highly infectious.

Figure 4: In comparison of nasal organoid titers and airway titers—were they from the same individual? Again, all copy number, no infectious virus.

The nasal and airway organoids were derived from different donors. Three independent experiments were performed using pairs of airway and nasal organoids selected randomly. Thus, the peak viral load varied among different experiments. Nonetheless, the findings suggested that nasal organoids were more susceptible to HRV-C and sustained a higher level of active viral replication than airway organoids.

Figure 4e: Have you got lower magnification images? Difficult to make out.

We didn't capture lower-magnification images. With these SEM figures, we intend to demonstrate the deformed cilia in HRV-C infected airway organoids, which were consistent with the confocal imaging results.

Line 176: No stats to back this up. What is the p-value?

We understand you mean Line 186. The p-value was calculated using Student's t-test. A "***" has been labeled on each bar in Fig. 4a, 4b, 4c to indicate significant differences. The replication data indicated that nasal organoids were more susceptible to HRV-C and sustained more active viral replication than airway organoids.

P-value of Fig. 4a 1000 copy/cell: 24h, 0.0035; 48h, 0.0010; 72h, 0.0004.

P-value of Fig. 4a 100 copy/cell: 24h, 0.1223; 48h, 3.9E-05; 72h, 1.83E-06.

P-value of Fig. 4b 1000 copy/cell: 24h, 0.002; 48h, 0.022; 72h, 0.003.

P-value of Fig. 4b 100 copy/cell: 24h, 0.008; 48h, 0.005; 72h, 1.9E-05.

P-value of Fig. 4c 1000 copy/cell: 24h, 8.64E-06; 48h, 0.001; 72h, 0.0003.

P-value of Fig. 4c 100 copy/cell: 24h, 0.02; 48h, 0.13; 72h, 0.0001.

P-value of Fig. 4c 10 copy/cell: 24h, 0.0004; 48h, 0.026; 72h, 0.059.

Figure 5g: Is that following treatment with CYT387?

The Poly(I:C) stimulation assay was performed in the airway and nasal organoids without any involvement of CYT387 treatment.

Figure 6: Not sure this aligns with their aims and probably needs further work to exclude other factors. Or move to supplementary as not specific to HRV but all viruses.

We agree with the reviewer's comment. This part of data is not specific to HRV-C, and seems redundant to the whole story. Thus, we decide to delete the related data.

Figure 7a-b: Need isotype or secondary-only control for the staining as this could be non-specific—especially for 7b. Like Figure 6, not sure how this all fits here. Should be included as part of the model characterization earlier on. Seems out of place here.

The Fig. 7 is Fig. 6 in the revised manuscript. The isotype IgG control was utilized for flow cytometry (Fig. 6a) and IF staining (Fig. 6b). Cells are gated and positivity is set based on the background staining of the isotype control in flow cytometry analysis. We have amended the figure legends accordingly. We also enclose the IF images with the Isotype IgG control to show the background signal as follows.

In this section, we presented the characterization of CDHR3 expression in organoids since it is directly related to the antibody-blocking experiment. We believe it is a rational presentation, as it provides the necessary context for understanding the results of the antibody-blocking experiment and elucidating the role of CDHR3 in HRV-C replication.

Figure 7c-d: This is useful and shows the potential application of the nasal organoids, which does not have the immunosuppressive agent.

We appreciate the reviewer's comment.

Figures and Tables

The figures and tables are mostly clear and informative but lack appropriate statistical tests. Some figures could be made bigger (1a and 2a diagram) and supplemental data necessary for understanding HRV-C clinical strain differences would be beneficial in the first figure.

We have made revisions to address the statistical issues in the manuscript.

The images in Fig. 1a and 2a have been enlarged accordingly.

Phylogenetic analysis of these HRV-C clinical strains has been shown in ED Fig. 1 and ED Fig. 2.

Strengths of the Manuscript

The manuscript provides a system for propagating HRV-C, overcoming a significant challenge in the field. Others have shown this 10+ years ago in differentiated cell culture models (<https://pubmed.ncbi.nlm.nih.gov/23035218/>, reference #8), which appears to be the same model used here. The uniqueness of this manuscript is the immunosuppression that enhances viral replication in airway cells (not nasal).

It offers a detailed comparison of immune responses between different types of respiratory organoids.

The study has some well-described procedures and results but details are missing as described above.

The implications for future research and antiviral development are justified.

We have addressed this issue in the prior context. Primary epithelial cells, including airway/bronchial epithelial cells, are generally not expandable; the limited primary cells might be sufficient for unsophisticated studies such as delineating virus infections. However, they hardly support more demanding studies, such as serial virus propagation and detailed characterization. That's why many human respiratory viruses remain uncultivable, despite the commercial availability of human airway/bronchial epithelial cells. The respiratory organoids provide a unique and robust model system, enabling us to conduct the challenging studies reported in the paper.

Major Weaknesses

Definition of 'organoid' needs to be stated early on.

We have introduced organoids in the revised manuscript on page 4.

Clarity of 'airway' definition and consistent terminology used.

An "airway" is a part of the respiratory system through which air flows, including the upper airway (the nasopharyngeal region) and lower airway (trachea, bronchi and bronchiole). The nasal organoids and airway organoids simulate the human upper and lower airway, respectively.

The study primarily focuses on HRV-C, and while it mentions the potential for studying other uncultivable viruses, it does not provide empirical evidence for this application.

The rationales for using respiratory organoids to propagate the previously uncultivable or poorly cultivable viruses have been specified in the introduction on page 3, 1) the respiratory tropism of HRV-C; 2) the ability of nasal and airway organoids to accurately simulate the native epithelium in human airways. Moreover, the respiratory organoid culture system enables us to maintain a stable and expandable source. The rationales apply to all viruses of human respiratory tropisms. Theoretically, nasal and airway organoids can sustain the propagation of all human respiratory viruses, including those previously uncultivable viruses.

What is the sequence of HRV? Has it undergone any mutation from replicating with/without immune suppression?

The genome sequence of isolated HRV-C3 was shown in the Supplementary Information file. We sequenced the viruses in the initial clinical specimen and the viruses after 1 and 6 consecutive passages in nasal organoids with immune suppression, and no adaptation mutation was identified.

Low replicate values, especially for early model validation (n=1 or 2), and therefore inappropriate statistical analysis applied.

In Fig. 1b, due to the limited volume of clinical specimens, we only inoculated each HRV-C clinical specimen into one well of organoids. As a result, only one reading is presented at each time point. The statistical analysis has been revised in the manuscript.

Minor Textual Errors

Page 4, Line 16: “in vivo characterization.” - Consider specifying the types of in vivo characterizations referred to.

Thanks for the reviewer’s comment, we have revised this in the revised manuscript on page 5.

Page 5, Line 38: “Moreover, virus growth exhibited significant variations” - Consider explaining the nature of these variations for clarity.

We have specified the factors contributable to the “significant variations” on page 3. The “significant variations” were noted in the publication that reported the first isolation of HRV-C using specimens from individuals with sinusitis⁷.

Noteworthy Results

The development of an organoid-based system to propagate human rhinovirus C (HRV-C) by CYT387-treated airway cell cultures or untreated nasal cell cultures.

Thanks for the reviewer’s comment.

Significance to the Field

The work is significant as it may provide a useful method for studying HRV-C and potentially other uncultivable viruses (though this claim isn’t strongly backed), despite being similar to previously established models.

We appreciate the reviewer’s comment. Again, we’d like to emphasize that this is the first report utilizing respiratory organoids for the propagation and characterization of previously poorly uncultivable virus HRV-C. This will be a landmark study for HRV-C research.

Comparison to Established Literature

The method resembles differentiated cell culture models used over a decade ago, but with the unique addition of immunosuppression enhancing viral replication in airway cells. Relevant reference: <https://pubmed.ncbi.nlm.nih.gov/23035218/>.

We have addressed the issue in the prior text.

Support for Conclusions and Claims

Some conclusions and claims need additional evidence, particularly regarding the statistical analysis and infection results.

We have addressed the relevant issues accordingly.

Flaws in Data Analysis and Interpretation

The data analysis has flaws, notably the inappropriate use of the Student's t-test with insufficient sample sizes, resulting in unreliable p-values.

We have revised the statistical analysis in the manuscript.

Methodology Soundness

The methodology has potential but lacks adequate detail in several areas, such as donor characteristics and precise cell source locations, which need clarification for reproducibility.

We have revised the methodology section accordingly, providing additional details.

Reproducibility Detail

Insufficient detail is provided in the methods for full reproducibility, specifically regarding donor characteristics, cell sources, and CYT387 catalog number.

We have revised the methodology section accordingly.

Reviewer #2 (Remarks to the Author):

General:

The authors have previously established the first human respiratory organoid culture system, which is a significant achievement and highly appreciated. Using this previously established system, the authors aimed to develop an organoid-based system to reproducibly propagate HRV-C and characterize virus-host interactions using respiratory organoids.

Major Comments:

1. The data indicate that HRV-C infects cells in both airway and nasal organoids. However, it is not clear whether the data fully support the goals stated by the authors. For instance, what advantages does the organoid-based system offer over the established Air-Liquid Interface (ALI) culture? The manuscript seems to underappreciate the precedent work using primary human bronchial epithelial cells in ALI culture, which has been instrumental in numerous

pioneering studies, including the discovery and characterization of HRV-C infection and related transcriptional and epigenetic studies.

Please see the following articles: PMID: 35570279, PMID: 33188283, PMID: 28472984, PMID: 38843491.

We appreciate the reviewer's comment.

We don't intend to underappreciate the importance of human primary epithelial cells. We acknowledge the crucial role of primary epithelial cells in HRV-C research. We also mentioned that the human primary epithelial cells had been used to study two unculturable viruses, bocavirus and coronavirus HKU1, in the introduction section on page 3.

Primary epithelial cells, including airway/bronchial epithelial cells, are generally not expandable, they are sufficient for unsophisticated studies such as delineating virus infections. However, they barely support more demanding studies, such as serial virus propagation and detailed characterization. That's why many human respiratory viruses remain uncultivable, although human airway/bronchial epithelial cells are commercially accessible or home-prepared in many labs. The previous publications of HRV-C have relied on either reverse genetic techniques or clinical specimens to obtain HRV-C viruses for studying the infection in human primary epithelial cells. Apparently, primary tissues and primary epithelial cells were unable to sustain serial propagation of HRV-C, except for an adopted strain C15a. The issues related to C15a are discussed on page 12 and 13.

In contrast, the respiratory organoids represent a unique and robust model system, in which expansion culture provides a stable and long-term expanding source (like routine cell culture), while differentiation protocols enable us to generate large amounts of physiologically active respiratory epithelial cells. Utilizing nasal and airway organoids, we isolated HRV-C directly from clinical specimens with high efficiency and then serially multiple HRV-C subtypes for detailed characterizations, which has never been achieved in the past two decades of HRV-C research. Overall, the key advantages of our organoid system over primary epithelial cell culture are long-term expandability and high stability.

2. Further validation and evidence are required for the conclusion that nasal organoids are more susceptible to HRV-C than airway organoids. Due to the very small sample size, the current data are insufficient to fully account for potential experimental or individual variability. Therefore, the generalization of this finding in comparison between the airway and nasal organoid is premature.

We have demonstrated that HRV-C replicated efficiently in nasal organoids even after inoculation at 1 viral gene copy/cell, whereas it did not do so in airway organoids (Fig. 4a, 4b, and 4c). We obtained highly consistent results from 3 pairs of randomly selected organoids derived from 6 different donors with three technical replicates. Flow cytometry analysis further revealed that HRV-C positive cells were more abundant in nasal organoids compared to airway organoids (Fig. 4d). Based on these findings, we believe it would be a solid conclusion that nasal organoids are more susceptible to HRV-C infection than airway organoids. Again, the

robust respiratory organoid culture system enabled us to conduct these detailed dissections, which is impossible to achieve in primary epithelial cell culture.

3. The manuscript speculates on the broader significance of the organoid system for uncultivable human and animal viruses without providing sufficient evidence. The same speculation could be made with ALI cultures of human BECs. Thus, additional data are required to support these claims and demonstrate why this system would be more appropriate than the ALI culture of HBECs.

The rationale for using respiratory organoids to propagate the previously uncultivable or poorly cultivable viruses has been specified in the introduction on page 3, 1) the respiratory tropism of HRV-C and other viruses; 2) the ability of nasal and airway organoids to accurately simulate the native epithelium in human airways. Moreover, the respiratory organoid culture system enables us to maintain a stable and expandable source. The rationales apply to all viruses of human respiratory tropisms. Theoretically, our nasal and airway organoids can sustain the propagation of all human respiratory viruses, including those previously uncultivable viruses.

As aforementioned, the key advantages of our organoid system over primary epithelial cell culture are long-term expandability and high stability. Primary epithelial cells only sustain limited passage and expansion, whereas organoids are stably expandable for over half a year. As the introduction on page 4, “In this two-phase organoid culture system, expansion culture provides a stable and long-term expanding source, while differentiation protocols enable us to generate large amounts of physiologically active respiratory epithelial cells. Thus, the respiratory organoid culture system allows us to rebuild and propagate the entire human respiratory epithelium in culture plates with excellent efficiency and stability”.

Minor Comments:

1. The legend and schema for Figure 1 need improvement. It is unclear whether they correspond to airway organoids, nasal organoids, or both. To improve the readability and comprehension of the data presented, clarity in figure legends and schematics is required.

Thanks for the reviewer's comment. The figure legends and schematics have been revised accordingly.

2. In both Fig 2d and 3e, ciliated cells are underneath the two cells marked by Phalloidin, which seems strange. It appears that the images are presented by maximum intensity projection. Including orthogonal views would help to indicate the infection of HRV-C in ciliated cells.

Thanks for the reviewer's comment. Orthogonal views have been added in the revised manuscript.

Reviewer #3 (Remarks to the Author):

References

- 1 Li, C. *et al.* Human airway and nasal organoids reveal escalating replicative fitness of SARS-CoV-2 emerging variants. *Proc Natl Acad Sci U S A* **120**, e2300376120 (2023). <https://doi.org/10.1073/pnas.2300376120>
- 2 Chiu, M. C. *et al.* A bipotential organoid model of respiratory epithelium recapitulates high infectivity of SARS-CoV-2 Omicron variant. *Cell Discov* **8**, 57 (2022). <https://doi.org/10.1038/s41421-022-00422-1>
- 3 Chiu, M. C. *et al.* Human Nasal Organoids Model SARS-CoV-2 Upper Respiratory Infection and Recapitulate the Differential Infectivity of Emerging Variants. *mBio* **13**, e0194422 (2022). <https://doi.org/10.1128/mbio.01944-22>
- 4 Zhou, J. *et al.* Differentiated human airway organoids to assess infectivity of emerging influenza virus. *Proc Natl Acad Sci U S A* **115**, 6822-6827 (2018). <https://doi.org/10.1073/pnas.1806308115>
- 5 Baggen, J., Thibaut, H. J., Strating, J. & van Kuppeveld, F. J. M. The life cycle of non-polio enteroviruses and how to target it. *Nat Rev Microbiol* **16**, 368-381 (2018). <https://doi.org/10.1038/s41579-018-0005-4>
- 6 Bochkov, Y. A. *et al.* Mutations in VP1 and 3A proteins improve binding and replication of rhinovirus C15 in HeLa-E8 cells. *Virology* **499**, 350-360 (2016). <https://doi.org/10.1016/j.virol.2016.09.025>
- 7 Bochkov, Y. A. *et al.* Molecular modeling, organ culture and reverse genetics for a newly identified human rhinovirus C. *Nat Med* **17**, 627-632 (2011). <https://doi.org/10.1038/nm.2358>

REVIEWERS' COMMENTS

Reviewer #1 (Remarks to the Author):

The authors have gone some way to addressing concerns, but the major issue is the reliance on copy number for viral load measurements. There is no attempt to quantify infectious virus, which would make this model very valuable to the community.

Reviewer #3 (Remarks to the Author):
